# Spatio-Temporal LLM: Reasoning about Environments and Actions

**Haozhen Zheng** [1]  **Beitong Tian** [1]  **Mingyuan Wu** [1]  **Zhenggang Tang** [1]  **Klara Nahrstedt** [1]  **Alexander Schwing** [1]

## Abstract

Despite significant recent progress of Multimodal Large Language Models (MLLMs), current MLLMs are challenged by "spatio-temporal" prompts, i.e., prompts that refer to 1) the entirety of an environment encoded in a point cloud that the MLLM should consider; and simultaneously also refer to 2) actions that happened in part of the environment and are encoded in a short egocentric video clip. However, such a holistic spatio-temporal understanding is important for agents operating in the real world. To address this challenge, we first develop a framework to collect a large-scale dataset. Using the collected "Reasoning about Environments and Actions" (REA) dataset, we show that recent MLLMs indeed struggle to correctly answer "spatio-temporal" prompts. Building on this dataset, we study two spatio-temporal LLM (STLLM) baselines: 1) STLLM-3D, which directly fuses point cloud, video, and text representations as inputs to the LLM; and 2) STLLM-Aligner, which aligns spatial context with video and text before LLM decoding. Both baselines aim to enhance spatial understanding of environments and temporal grounding of egocentric observations. On REA, the STLLM baselines outperform existing models, demonstrating the effectiveness of our designs. Code and data are available at https://zoezheng126.github.io/STLLM-website/.

## 1. Introduction

Multimodal Large Language Models (MLLMs) have been remarkably successful in the last few years, and have gained some spatio-temporal reasoning proficiency. To achieve this, recent efforts by Zhu et al. (2025) utilize 3D positional embeddings to enhance the 2D features with spatial context. Further, Liu et al. (2025); Liu et al. (2024a); Li et al. (2024) have enabled models to reason beyond 2D images and toward richer spatial representations by extending existing datasets to incorporate spatial data. Man et al. (2024); Huang et al. (2024); Ma et al. (2023) augment MLLMs with additional 3D representations, such as depth maps or point clouds, as auxiliary input channels to provide complementary structural cues beyond 2D RGB images. Wu et al. (2025a) further incorporate a VGGT (Wang et al., 2025a) encoder to transfer pretrained 3D geometric cues into the MLLMs. Despite these advances, existing models have a shortcoming: they don't reason about spatial context beyond what is provided in video clips for two reasons: 1) data that provides both a global spatial context and locally confined temporal observations is not available; and 2) they augment temporal observations with spatial information rather than treating temporal and spatial data as different modalities.

Yet, reasoning about an unfolding event observed from an egocentric perspective, while anchoring it in a broader world context is crucial for wearable AI, embodied AI, and spatio-temporal question answering. For example, robots interacting in the real world must interpret observations not only by recognizing the current action, but by situating it within the surrounding context. Relying solely on egocentric perception yields rich local detail but prevents the agent from forming a coherent map of the environment. In contrast, purely allocentric localization provides global structure but discards the fine-grained, first-person evidence needed to interpret ongoing interactions. Our goal is to align both perspectives, enabling the model to ground local observations within a consistent world-centric representation.

For this, we first develop a Spatio-Temporal Understanding Question Answering (QA) data collection pipeline. It retrieves both egocentric spatially local video clips and a global scene representation. Using our pipeline, we collect the **"Reasoning about Environments and Actions" (REA) data** shown in Fig. 1: it includes five tasks—relative direction, relative distance, find-my-item, furniture affordance prediction, action planning—each designed to test different aspects of spatio-temporal reasoning. We also study two **"Spatio-Temporal LLM" (STLLM)** baselines which treat temporal and spatial data as different modalities.

On REA data, we show: 1) spatio-temporal understanding

---

[1]University of Illinois Urbana-Champaign. Correspondence to: Haozhen Zheng <haozhen3@illinois.edu>.

*Proceedings of the $43^{rd}$ International Conference on Machine Learning*, Seoul, South Korea. PMLR 306, 2026. Copyright 2026 by the author(s).

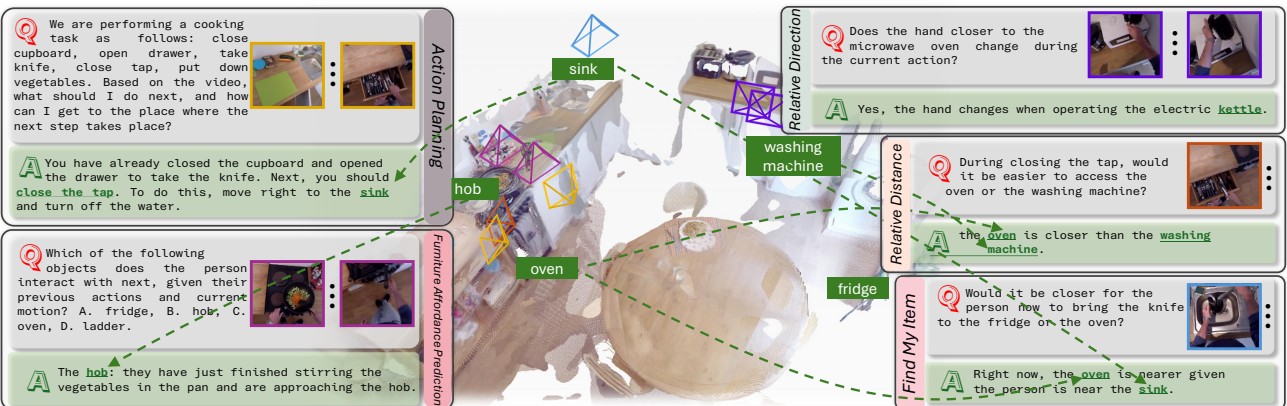

*Figure 1.* Spatial *and* temporal reasoning is needed to answer prompts in "Reasoning about Environments and Actions" (REA). Ego-centric videos only show part of the point cloud environment.

remains a challenge for current MLLMs, as existing models achieve an overall `ChatGPT-4o` (OpenAI, 2024) LLM Judge accuracy of only 23.85% to 31.46% across tasks; and 2) our STLLM baselines reach 41.89% overall accuracy and 47.32% average categorical accuracy, highlighting that temporal and spatial modalities are important.

In summary, our contributions are as follows: 1) we develop a dataset collection pipeline and collect REA (see Sec. 3 for more); and 2) we study STLLM baselines to enhance spatio-temporal understanding in language models (see Sec. 4 for more). Code and data will be released.

## 2. Related Work

**Image LLM.** Recent commercial VLMs (OpenAI, 2024; Meta, 2024; Anthropic, 2024) have demonstrated strong results across a range of image-language tasks, including image understanding and chart and task-based question answering. To mimic, most modern, open-source image-language models (Deitke et al., 2025; Bai et al., 2023; Wang et al., 2024a; Bai et al., 2025; Shi et al., 2025; Li et al., 2025; Tong et al., 2024) adopt a common architecture: an image encoder, a connector module that pools and projects visual features into the VLM's embedding space, and a language decoder. Post-training techniques such as visual instruction tuning (Liu et al., 2023) are often applied to further enhance these models' ability to understand and follow natural language instructions.

**Video LLM.** Early video-language models (Zhang et al., 2023; Cheng et al., 2024b) adopt the image LLMs architecture: encode video frame features and connect them to an instruction-tuned language model via a projection layer. Recent efforts pursue long-form videos (Wu et al., 2025b; Chen et al., 2025; Xu et al., 2025), improve streaming efficiency for real-time applications (Qian et al., 2024; Zhang et al., 2025b; Chen et al., 2024b), and introduce memory to enable effective long-term grounding and downstream

question answering (Wang et al., 2024b; Mangalam et al., 2024).

**3D LLMs.** Recent work integrates 3D information into LLMs to enable spatially grounded reasoning, including object grounding, scene description, and path planning, through different combinations of visual inputs and 3D alignment mechanisms. **(i) Multi-view image + 3D.** Many approaches extract 2D features from multiple views and project them into a shared 3D representation, or encode point clouds with pretrained 3D encoders and align them with the LLM embedding space for spatial reasoning (Hong et al., 2023; Chen et al., 2024a). These methods primarily rely on aggregating static views to construct allocentric spatial representations. **(ii) Video + 3D.** Extending this paradigm, recent models combine egocentric video with 3D geometry, often using per-frame depth maps or point-based reconstructions to align video observations with 3D space (Zheng et al., 2025; Zhu et al., 2025; Wu et al., 2025a). **(iii) Single image + 3D.** Another line of work interleaves a single egocentric image with a 3D scene representation to support situational awareness, enabling the model to describe and reason about the environment from the perspective of the image input (Linghu et al., 2024; Huang et al., 2024). However, none of these works are naturally suited for answering REA, as it requires understanding action-aware motion and aligning observations across an egocentric video and an allocentric point cloud at the same time.

**Structured Spatial Reasoning.** Another stream of works reason about spatial relations in 3D environments by introducing explicit scene structures as intermediate representations. These approaches use VLMs to construct symbolic or programmatic representations, such as scene graphs (Gu et al., 2024; Hsu et al., 2023) or differentiable logic programs (Huang et al., 2025), and perform reasoning over these external structures rather than learning spatial representations end-to-end. While effective for structured reasoning, these methods primarily rely on externally imposed

scene abstractions and do not directly improve the VLM's internal spatial understanding.

**Data for spatial understanding.** Several recent benchmarks evaluate spatial understanding in multimodal vision-language models (Yeh et al., 2025; Fu et al., 2025; Yang et al., 2025a; Zhang et al., 2025c; Cheng et al., 2024a). These datasets primarily target object-centric spatial reasoning over static images or short video clips, focusing on relationships such as relative position and layout. Benchmarks by Ji et al. (2025) and Yang et al. (2025b) also involve spatial reasoning in 3D environments, but are centered on evaluating Vision-Language-Action models through downstream embodied tasks, with an emphasis on task execution and policy performance. However, current benchmarks do not fully reflect the inherent complexity of real-world inputs, which naturally involve dynamic actions and multi-perspective information. This motivates REA: a dataset designed to evaluate action-aware spatio-temporal reasoning with implicit egocentric-allocentric alignment. We provide a detailed dataset comparison in Appendix H.2.

# 3. Reasoning about Environments and Actions (REA)

Our goal: equip MLLMs with spatial *and* temporal understanding. As shown in Fig. 1, the model should answer prompts that require 1) spatial understanding about a *global* 3D environment, represented

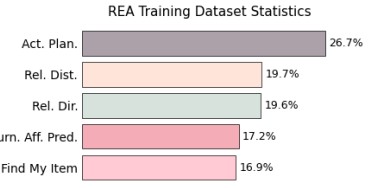

*Figure 2.* Training data statistics.

via a point cloud; and 2) *local* temporal understanding, represented via an egocentric video that covers only part of the environment. For this, we first develop a dataset collection pipeline benefitting from existing data: 1) dense action annotations from EPIC-KITCHENS (Damen et al., 2018); 2) object segmentation annotations by VISOR (Darkhalil et al., 2022); and 3) sparse point clouds from EPIC-FIELDS (Tschernezki et al., 2023). Using the pipeline, we collect the **"Reasoning about Environments and Actions" (REA)** data: question-answer pairs partitioned into five tasks that can be used to equip a MLLM with spatio-temporal understanding. Answering prompts requires to understand global scene context from a 3D scene and localized temporal cues from an egocentric video. Next, we first introduce the five tasks that form the REA dataset as shown in Fig. 2, followed by details of the data collection pipeline (Sec. 3.1). Detailed task definitions are provided in Appendix A.

**Relative Direction.** This task evaluates whether the model can infer changes in an object's relative direction (left-

/right/front/back) with respect to the person across actions, based on egocentric video and global 3D context.

**Relative Distance.** This task assesses whether the model can reason about changes in the person's proximity to objects over time, including both absolute distance changes and comparative distances between objects.

**Find My Item.** This task tests object localization and spatial navigation by requiring the model to identify where an object was left and infer how to reach it using scene-level spatial context.

**Furniture Affordance Prediction.** This task evaluates the ability to predict which nearby furniture object the person is likely to interact with next, based on recent actions, motion trends, and surrounding layout.

**Action Planning.** This task measures whether the model can anticipate the next action in a task sequence and generate a navigation instruction to reach the location of the upcoming interaction.

## 3.1. Data Collection Pipeline

As illustrated in Fig. 3, we construct a six-step data collection pipeline that integrates egocentric video, 3D geometry, and action annotations to generate spatio-temporal question-answer pairs. We use this pipeline to obtain both question-answer pairs (Sec. 3.1.1) and point cloud representations (Sec. 3.1.2). Manual quality control is performed to ensure data reliability (Appendix B.1). The following sections describe each step in detail.

### 3.1.1. QUESTION-ANSWER (QA) GENERATION

To generate diverse question-answer pairs, we start with 3–5 question templates for each task which are rephrased via an LLM upon completion. To ensure that answering a question needs spatio-temporal understanding, completing a template requires the use of four steps: 1) query video sampling; 2) 3D position estimation; 3) spatial relationship estimation; and 4) navigation movement estimation.

**1) Query Video Sampling.** We begin by sampling 20-40 second clips, which serve as the visual input to the MLLM during training or inference. Clips are sampled from longer recordings in the EPIC-KITCHENS (Damen et al., 2018) data. Benefiting from the dense action annotations provided by EPIC-KITCHENS (Damen et al., 2018), we partition a sampled clip into a series of fine-grained action intervals, each approximately 3-5 seconds long. We then sample action intervals from the clip. To be eligible, the selected actions must meet task-specific criteria. E.g., for *Relative Direction* and *Relative Distance*, we select two non-consecutive actions from the clip with a sufficiently long interval between them, ensuring a high likelihood of a

*Figure 3.* **Dataset generation pipeline.** Note, in 2&3), camera poses (in green), sampled across the action interval, are used to compute the relative direction and distance between the person (moving along the arrow) and the object. To obtain per-frame camera poses for the query video, we first use VGGT (Wang et al., 2025a) to re-compute the point cloud (step 5) and subsequently apply Reloc3r (Dong et al., 2025) (step 6).

shift in relative angle or distance between the query object and the person. In contrast, for *Furniture Affordance* Prediction, where the goal is to infer which piece of furniture the person will interact with, the query actions are selected as the next action following the video clip. This ensures that the ground-truth interaction occurs after the clip ends, allowing the MLLM to predict the affordance without directly observing the final interaction.

**2) 3D Position Estimation.** Next, we compute the 3D locations of both the person and query objects. Given the short duration of each action interval and the typically small movements in kitchen scenarios, we assume that the person's location does not change significantly during a single action. To estimate the person's location during an action, we leverage the sparse image registration provided by EPIC-FIELD (Tschernezki et al., 2023) and use the mean camera pose of the registered frames within the action interval as a proxy for the person's 3D location.

To determine the 3D location of a query object, we utilize the 2D segmentation masks from VISOR (Darkhalil et al., 2022), along with the estimated human pose at the moment the person is interacting with the object. Assuming the object remains in close proximity to the person during the interaction, we project the 2D segmentation mask onto the COLMAP (Schönberger & Frahm, 2016) point cloud and compute the average 3D position of the projected points using a frame near the middle of the interaction. This average serves as the estimated 3D location of the query object.

**3) Spatial Relationship Estimation.** After estimating the ground-truth 3D poses of the person and the objects, we compute the spatial relationship between the object and the person's movement as observed in the query video. To simplify the analysis, we constrain the object to remain stationary throughout the movement by ensuring that the person interacts with the object at most once.

For the *Relative Direction* task, computations are performed in the person's camera coordinate frame. By transforming the object's 3D location from world coordinates into the person's egocentric frame, we determine whether the object is positioned to the left, right, front, or back of the person.

For the *Relative Distance* task, we calculate the change in distance between the person and the object over time using their world-coordinate poses. Specifically, we sample five poses of the person across the action interval, compute the L2 distances to the object in world coordinates, and fit a linear regression to these values. A positive slope indicates the person is moving away from the object, while a negative slope indicates movement toward it. We apply a threshold of $\pm 0.05$ to classify whether the person is moving relative to the object or remains relatively stationary.

**4) Navigation Movement Estimation.** For *Action Planning*, we estimate navigational movement in world coordinates by thresholding displacement (1.5 m) to identify meaningful motion and computing the relative direction toward the destination. We then refine this preliminary estimate using a VideoLLM (Zhang et al., 2024), which observes the reference navigation video and adjusts the direction to better reflect the person's true navigational behavior.

### 3.1.2. POINT CLOUD RECONSTRUCTION

EPIC-FIELDS (Tschernezki et al., 2023) provides only sparse COLMAP (Schönberger & Frahm, 2016) reconstructions, which are too sparse to capture fine-grained geometry and to support accurate registration of dense video frames. Since per-frame camera poses are essential for learning 2D–3D alignment, we therefore obtain dense camera poses using a two-step approach: reconstructing a dense point cloud with pose-free methods (Tang et al., 2025; Wang et al., 2025a), followed by registering video frames to the point cloud using Reloc3r (Dong et al., 2025).

**5) Point Cloud Reconstruction.** To reconstruct dense point clouds from egocentric videos, we select hand-free frames using Grounded SAM2 (Jiang et al., 2024; Kirillov et al., 2023; Ren et al., 2024b;a; Liu et al., 2024b) and choose a sparse set of representative views that provide broad scene coverage. Specifically, we apply K-means clustering on camera poses to select 25 frames per recording for recon-

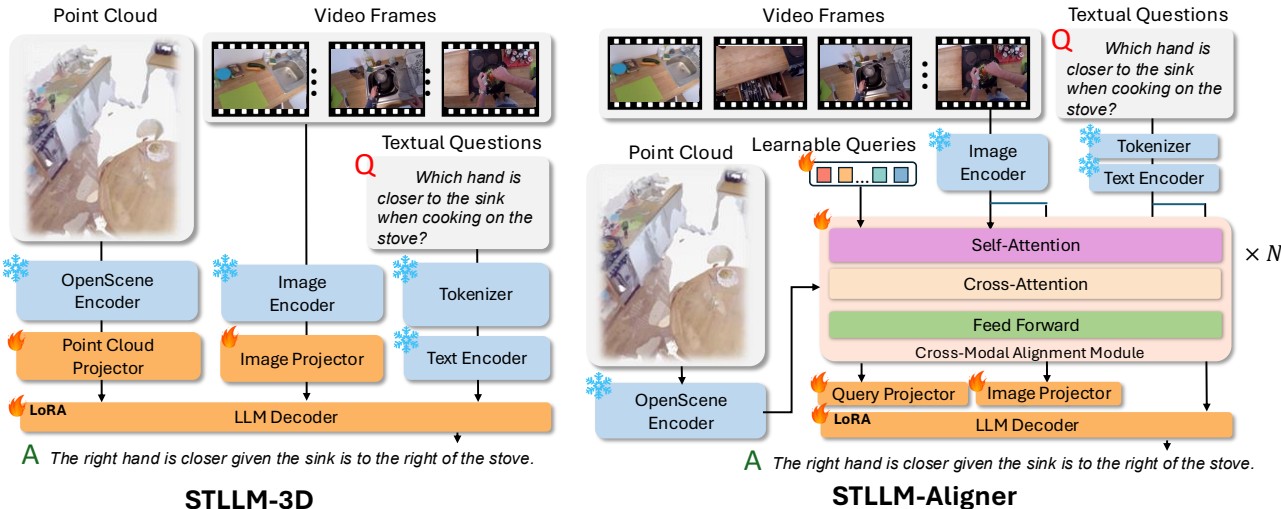

*Figure 4.* Architectures of STLLM-3D and STLLM-Aligner.

struction. All reconstructed point clouds are manually verified for quality.

**6) Frame to Point Cloud Registration.** To obtain per-frame camera poses, we follow Reloc3r (Dong et al., 2025). For each scene, we build a database of the 25 reconstructed frames and their camera poses obtained from VGGT (Wang et al., 2025a). Given a new video frame, Reloc3r retrieves nearby reference images and estimates relative poses to register the frame to the reconstructed point cloud.

## 4. Spatio-Temporal LLMs (STLLMs)

To handle spatio-temporal reasoning, we want spatio-temporal LLMs to process a global 3D point cloud $\mathbf{P}$, a video $\mathbf{V}$, a textual instruction $\mathbf{T}$, and camera parameters for each video frame, including intrinsics and extrinsics. Importantly, the point cloud $\mathbf{P}$ offers a *global* 3D context, and the video input $\mathbf{V}$ records egocentric human actions situated locally within this environment. Concretely, the point cloud is defined as $\mathbf{P} = \{[\mathbf{p}_{xyz}, \mathbf{f}_{rgb}]\} \in \mathbb{R}^{N \times 6}$, where $\mathbf{p}_{xyz} \in \mathbb{R}^3$ are the 3D coordinates of each point and $\mathbf{f}_{rgb} \in \mathbb{R}^3$ are the corresponding RGB colors. The video is represented as a set of $T$ image frames $\mathbf{V} = \{\mathbf{I}_1, \mathbf{I}_2, \ldots, \mathbf{I}_T\}$, where each frame $\mathbf{I}_t \in \mathbb{R}^{H \times W \times 3}$ is associated with its own intrinsic and extrinsic matrices $\mathbf{K}_t \in \mathbb{R}^{4 \times 4}$ and $\mathbf{E}_t \in \mathbb{R}^{4 \times 4}$.

To treat temporal and spatial data as different modalities, we assess two baselines: 1) **STLLM-3D** directly concatenates along the sequence dimension 3D point cloud, video, and text inputs for decoding; 2) **STLLM-Aligner** extracts spatial queries via a cross-modal alignment module, which results in a compact spatial representation that is used as LLM input. Both baselines aim to reason about global scene context while following fine-grained temporal dynam-

ics. Architectures of STLLM-3D and STLLM-Aligner are illustrated in Fig. 4. Both baselines are based on LLaVA-Video-Qwen2 (Zhang et al., 2024), but extend it to handle 3D spatial information in addition to video and text. Both share the same vision and point cloud encoder. For the vision encoder, following LLaVA-Video-Qwen2 (Zhang et al., 2024), we adopt SigLip (Zhai et al., 2023) as our vision encoder. For the point cloud encoder, given a dense point cloud $\mathbf{P}$, we first apply voxel-based downsampling to obtain a reduced set of representative points $\tilde{\mathbf{P}}$ before extracting point-wise feature embeddings $\mathbf{f}_{pcd}$ using a masked transformer decoder $\mathcal{T}_{pcd}$, i.e., $\mathbf{f}_{pcd} = \mathcal{T}_{pcd}(\tilde{\mathbf{P}}) \in \mathbb{R}^{N \times 768}$ (Peng et al., 2023). See Appendix E for training details.

**STLLM-3D.** The STLLM-3D baseline directly integrates 3D information with video and text. After extracting point-wise features $\mathbf{f}_{pcd}$, we apply Farthest Point Sampling and grouping to form a set of compact spatial features. These features are projected into the language space via an MLP layer and concatenated with the image and text embeddings, which are then fed directly into the LLM decoder. Beneficially, STLLM-3D adopts a straightforward design that concatenates spatial, visual, and textual features for direct decoding. This simplicity makes the architecture easy to implement and computationally lightweight. However, given large and complex scenes, more token embeddings are required, which increases the LLM input and its computational cost. In other words, while STLLM-3D is effective at direct integration, it struggles with scalability. This limitation motivates the design of STLLM-Aligner, which seeks to compress spatial features into a compact representation.

**STLLM-Aligner.** The STLLM-Aligner baseline introduces a cross-modal alignment module to bridge frozen pointcloud features, video, and text. A set of learnable queries are used

*Table 1.* Comparison of models on various evaluation metrics. Sim = Sentence Similarity.

| Model | Sim (%) ↑ | CIDEr ↑ | BLEU (%) ↑ | METEOR (%) ↑ | ROUGE (%) ↑ |
|---|---|---|---|---|---|
| LLaVA-Video-7B-Qwen2 (Zhang et al., 2024) | 65.83 | 20.79 | 10.25 | 19.68 | 23.71 |
| LLaVA-OV-Qwen2-7B (Li et al., 2024) | 64.51 | 3.34 | 11.22 | 19.53 | 23.84 |
| Qwen2-VL-7B-Instruct (Wang et al., 2024a) | 52.99 | 35.08 | 19.11 | 17.38 | 25.37 |
| VideoLLaMa3 (Zhang et al., 2025a) | 39.14 | 10.85 | 2.15 | 7.85 | 14.12 |
| *Finetuned on REA dataset* | | | | | |
| LLaVA-Video-7B-Qwen2[†] | 85.26 | 387.72 | 60.34 | 43.18 | **72.09** |
| LLaVA-OneVision-Qwen-7B[†] | 85.09 | 400.23 | 61.90 | 42.41 | 71.11 |
| STLLM-Aligner[‡] (w Pos.Enc.) | 71.34 | 170.63 | 39.46 | 28.54 | 50.02 |
| STLLM-Aligner (w Pos. Enc.) | 85.70 | 406.54 | 61.90 | **44.16** | **72.09** |
| **STLLM-Aligner** | 85.58 | **406.68** | **62.01** | 43.94 | 72.03 |
| **STLLM-3D** | **85.99** | 405.48 | 61.99 | 44.04 | 72.07 |

*Note.* [†] indicates the existing model is finetuned on our REA dataset. [‡] LLM layers are not finetuned.

to attend to image and text features, and spatial context from the point cloud is incorporated through cross-attention. The resulting spatial queries, together with the image and text embeddings, are passed into the LLM decoder. This design provides a compact yet informative representation of the 3D scene. However, the token compression introduced by the alignment mechanism is harder to interpret. Hence, STLLM-Aligner trades the efficiency challenge of STLLM-3D with its own limitations. To understand the trade-offs, we study both baselines.

For the STLLM-Aligner, we also study use of a high-frequency positional encoding in the alignment module as a complementary enhancement. The positional encoding provides geometric cues beyond raw coordinates. Concretely, for each video frame, we back-project pixels using camera intrinsics and extrinsics to obtain per-pixel ray directions, which are then normalized and downsampled to match the patch-level tokens from the vision encoder. For the point cloud, each 3D point is transformed into the first camera frame, and its normalized direction from the camera origin is taken as its ray vector. These ray directions are then mapped through a high-frequency encoding function and projected with a lightweight MLP to align dimensions with the modality features. The resulting position-aware embeddings are fused with the original image and point cloud feature before entering the alignment module. This design complements the frozen encoders with explicit geometric cues.

## 5. Experiments

We now examine the effectiveness of our dataset and the baselines: 1) We compare to several state-of-the-art VideoLLMs, including the base model we pretrained from. For a fair comparison, we feed the global scene context to existing models in a multi-view image format, as they cannot directly process point cloud inputs. 2) We evaluate using standard

question answering metrics and two LLM-Judges (Zheng et al., 2023) to ensure consistency in reasoning and correctness.

**Dataset Statistics.** We construct the **REA** dataset using our proposed data generation pipeline. After manual validation, we obtain 24,371 training samples and 1,757 validation samples. The dataset inherits the action classes and over 300 annotated objects from EPIC-KITCHENS (Damen et al., 2018) and EPIC-FIELDS (Tschernezki et al., 2023), covering a wide range of kitchen activities. It features strong long-tail distributions in both training and test splits (e.g., 4,759 unique actions in training with over 20% appearing only once), highlighting its richness and diversity. This diversity is in part a consequence of the kitchen environment, which, as a common everyday setting, contains a wide variety of object categories and supports frequent human-object interactions. As a result, kitchens serve as a natural and challenging starting point for spatio-temporal visual question answering.

**Metrics.** We adopt standard QA and captioning metrics, including SenSim (Reimers & Gurevych, 2019), CIDEr (Vedantam et al., 2015), BLEU-4 (Papineni et al., 2002), METEOR (Banerjee & Lavie, 2005), and ROUGE-L (Lin, 2004). We additionally use LLM-based judges (Zheng et al., 2023) to assess semantic correctness for open-ended answers. Specifically, we use `ChatGPT-4o` (C) (OpenAI, 2024) and `Gemini 2.0 Flash` (G) (Google DeepMind, 2025) to compare model predictions with ground-truth answers and determine whether they convey the same underlying meaning. Task-oriented prompts instruct the judges to focus on semantic equivalence rather than exact string matching (Appendix C). Judge reliability is discussed in Appendix B.2.

**Model Implementation.** STLLM-3D and STLLM-Aligner baselines are built upon LLaVA-Video-7B-Qwen2 (Zhang

*Table 2.* LLM-Judge accuracy (%, higher is better, C = `ChatGPT-4o`, G = `Gemini 2.0 Flash`).

| Model | | Rel. Dir. | Rel. Dist. | Find My Item | Affordance | Action Plan. | Overall / Avg. |
|---|---|---|---|---|---|---|---|
| LLaVA-Video-7B-Qwen2 (Zhang et al., 2024) | C | 36.67 | 43.00 | 28.06 | 53.05 | 13.17 | 30.96 / 34.79 |
| | G | 46.00 | 42.67 | 38.49 | 56.27 | 27.33 | 39.50 / 42.15 |
| LLaVA-OV-Qwen2-7B (Li et al., 2024) | C | 15.33 | 36.00 | 25.54 | 50.18 | 9.00 | 23.85 / 27.21 |
| | G | 36.67 | 40.00 | 40.65 | 51.61 | 23.50 | 35.74 / 38.49 |
| Qwen2-VL-7B-Instruct (Wang et al., 2024a) | C | 38.33 | 9.67 | 15.47 | 40.50 | 15.00 | 24.38 / 23.68 |
| | G | 36.67 | 10.00 | 23.02 | 41.22 | **33.67** | 29.94 / 27.90 |
| VideoLLaMa3 (Zhang et al., 2025a) | C | **57.00** | 42.00 | 20.86 | 39.43 | 10.00 | 31.46 / 35.86 |
| | G | **70.33** | 38.33 | 42.45 | 39.07 | 13.33 | 36.03 / 40.70 |
| Qwen2.5-VL-7B (Bai et al., 2025) | C | 33.00 | 40.67 | 16.91 | 49.82 | 14.17 | 28.00 / 30.91 |
| | G | 48.67 | 34.33 | 29.14 | 50.54 | 21.83 | 34.26 / 36.90 |
| Qwen3-VL-8B (Team, 2025) | C | 29.67 | 32.33 | 14.39 | 26.88 | 16.67 | 22.82 / 23.99 |
| | G | 40.00 | 31.33 | 34.89 | 50.18 | **47.83** | 42.00 / 40.85 |
| InternVL3.5-8B (Wang et al., 2025b) | C | 28.00 | 47.33 | 28.06 | 28.67 | 15.00 | 26.98 / 29.41 |
| | G | 45.00 | 49.33 | 44.60 | 35.13 | 23.83 | 36.88 / 39.58 |
| *Finetuned on REA dataset* | | | | | | | |
| LLaVA-Video-7B-Qwen2[†] | C | 40.67 | 61.00 | 36.69 | 61.65 | 11.83 | 36.99 / 42.37 |
| | G | 44.00 | 61.00 | 56.12 | 57.35 | 20.50 | 42.92 / 47.79 |
| LLaVA-OV-Qwen-7B[†] | C | 41.00 | 66.00 | 32.73 | 59.86 | 15.33 | 38.19 / 42.98 |
| | G | 47.00 | 66.00 | 47.12 | 55.91 | 23.50 | 43.65 / 47.91 |
| STLLM-Aligner[‡] (w Pos.Enc.) | C | 56.67 | 49.67 | 28.78 | 63.80 | 7.50 | 35.38 / 41.26 |
| | G | 39.67 | 48.33 | 48.20 | 55.91 | 14.17 | 36.37 / 41.26 |
| STLLM-Aligner (w Pos.Enc.) | C | 49.00 | 69.00 | **38.13** | 59.50 | **17.00** | 41.43 / 46.53 |
| | G | 50.00 | 69.00 | 55.40 | 53.41 | 24.33 | 45.87 / 50.43 |
| **STLLM-Aligner** | C | 50.67 | **70.67** | 36.69 | 62.72 | 15.83 | **41.89 / 47.32** |
| | G | 51.33 | **70.67** | 55.04 | 55.56 | 23.83 | **46.50** / 51.29 |
| **STLLM-3D** | C | 48.00 | 68.00 | 35.61 | **65.69** | 14.83 | 40.94 / 46.43 |
| | G | 51.00 | 68.00 | **56.47** | **58.06** | 23.17 | 46.39 / **51.34** |

et al., 2024). They are finetuned on our REA dataset for one epoch using AdamW (Loshchilov & Hutter, 2019) with a cosine learning rate scheduler and a max learning rate of 1e−4 (training details in Appendix E). Finetuning is conducted on four NVIDIA H200 GPUs.

**Results on Standard QA Metrics.** Table 1 reports sentence-level and n-gram metrics. Despite straightforward handling of scene and video data, STLLM baselines improve upon existing models, including those finetuned on REA. This shows: spatio-temporal reasoning is yet unsolved.

**Results on LLM-Judge.** Table 2 reports LLM-Judge accuracy across the five tasks of the REA test set. As shown, off-the-shelf MLLMs are challenged by spatio-temporal reasoning. Earlier video and image-video baselines remain below 31.46%/39.50% overall accuracy under the ChatGPT-4o/Gemini judges. We further evaluate stronger open-source MLLMs, including Qwen2.5-VL-7B (Bai et al., 2025), Qwen3-VL-8B (Team, 2025), and InternVL3.5-8B (Wang et al., 2025b). These models improve on some individual tasks but still do not solve REA. For example, Qwen3-VL-8B (Team, 2025) reaches **42.00%** overall accuracy under the Gemini judge, largely due to its strong result on *Action Planning*, but this improvement is concentrated in a single task rather than consistent across REA. Similarly,

InternVL3.5-8B performs competitively on *Relative Distance* and *Find My Item*, but does not show consistent gains across tasks. Notably, Qwen3-VL-8B (Team, 2025) and InternVL3.5-8B (Wang et al., 2025b) are 8B-scale models, whereas our STLLM baselines are built on a 7B backbone, suggesting that the improvements of STLLM are not simply due to model scale.

Interestingly, VideoLLaMA3 (Zhang et al., 2025a) achieves superior performance on the *Relative Direction* task, likely due to exposure to spatially-grounded data during pre-training, but it does not show consistent advantages on the other tasks. While finetuning on REA brings noticeable gains for all models, our STLLM baselines achieve higher performance. In particular, STLLM-Aligner attains **41.89%/46.50%**, outperforming the directly finetuned LLaVA-Video-7B-Qwen[†] (**36.99%/42.92%**) and remaining stronger than the additional open-source baselines overall. This shows that developing spatio-temporal LLM architectures is beneficial and can yield gains over both off-the-shelf and REA-finetuned 2D counterparts.

Comparing the two STLLM variants further reveals a favorable computation–accuracy trade-off. STLLM-Aligner yields marginal gains over STLLM-3D under the ChatGPT-4o judge, while Gemini reports nearly identical performance.

*Table 3.* SQA3D Test Set - Correct Rate (%) per Question Type (GPT / Gemini).

| Model | What | Is | How | Can | Which | Other | Average |
|---|---|---|---|---|---|---|---|
| LLaVA-Video-7B-Qwen2 (Zhang et al., 2024) | 41.68 / 45.34 | 53.66 / 52.61 | 20.62 / 22.16 | 53.29 / 49.70 | 40.12 / 48.84 | 46.61 / 47.88 | 43.04 / 45.04 |
| LLaVA-OV-Qwen2-7B (Li et al., 2024) | 39.49 / 44.79 | 59.93 / 59.93 | 25.26 / 31.44 | 49.70 / 53.89 | 36.05 / 41.86 | 52.12 / 57.20 | 43.98 / 41.86 |
| Qwen2-VL-7B-Instruct (Wang et al., 2024a) | 33.82 / 34.37 | 51.92 / 51.92 | 21.65 / 17.53 | 52.69 / 49.70 | 38.37 / 41.28 | 50.00 / 50.85 | 40.42 / 40.24 |
| VideoLLaMa3 (Zhang et al., 2025a) | 38.76 / 39.85 | 56.10 / 63.07 | 39.69 / 38.66 | 47.31 / 55.69 | 33.14 / 34.88 | 52.54 / 54.66 | 44.29 / 47.16 |
| *Finetuned on REA dataset* | | | | | | | |
| LLaVA-Video-7B-Qwen† | 48.26 / 50.82 | 65.51 / 65.51 | 45.36 / 46.91 | **60.48** / 61.08 | 44.19 / 40.70 | 49.58 / 51.27 | 52.03 / 53.03 |
| LLaVA-OV-Qwen2-7B† | 38.76 / 47.53 | 56.10 / 62.72 | 39.69 / 39.18 | 47.31 / 49.10 | 33.14 / 40.12 | 52.54 / 50.00 | 44.29 / 48.97 |
| STLLM-Aligner‡ | 46.25 / 44.79 | 56.79 / 62.37 | 47.42 / 50.52 | 52.10 / **73.05** | **47.09** / 39.53 | 47.03 / 50.00 | 49.10 / 51.78 |
| STLLM-Aligner (w Pos. Enc.) | 49.17 / 50.55 | 63.93 / 65.51 | 51.31 / 50.00 | 56.02 / 58.08 | 46.20 / 45.35 | 55.17 / **58.90** | 53.32 / 54.62 |
| **STLLM-Aligner** | 49.73 / 51.74 | 65.16 / 64.46 | 50.52 / 48.97 | 58.68 / 58.08 | 50.00 / **50.00** | 55.51 / 55.51 | 54.40 / 54.71 |
| **STLLM-3D** | **50.27** / 51.92 | **66.55** / 66.90 | **53.09** / 52.06 | **60.48** / 62.28 | 46.51 / 43.02 | **57.20** / 58.05 | **55.21** / 55.65 |

*Note.* The models do not generate exact-match answers by design. We incorporate LLM judges for evaluation, where an answer is considered correct if it expresses the same meaning as the ground truth.

Since STLLM-Aligner introduces an additional cross-modal alignment module and both variants use comparable decoder sequence lengths, STLLM-3D provides a simpler and more computationally efficient alternative with competitive accuracy. We also assess use of additional positional encoding in the alignment module, but find little performance difference, likely because REA emphasizes question answering rather than explicit 3D grounding. Hence, positional encodings offer limited benefits. We provide additional analyses in Appendix F.

**Task difficulty.** Tasks such as *Find My Item* and *Action Planning* are generally more challenging, as they require open-ended answers and involve both spatial and temporal reasoning. Meanwhile, both *Relative Direction* and *Relative Distance* demand strong spatial understanding. Our STLLM baselines demonstrate superior overall performance with well-balanced results across tasks, highlighting their robust spatial-temporal reasoning capabilities.

**Modality Integrations.** To demonstrate that our models genuinely reason over space and time, we show that performance gains arise specifically from incorporating true 3D spatial structure. Replacing the point-cloud input with a pure 2D multi-view baseline leads to a clear drop in spatio-temporal QA performance (41.89% to 36.99% overall accuracy; 47.32% to 42.37% average accuracy in Table 2), indicating that our models benefit from reasoning over explicit 3D geometry rather than relying on frame-level appearance cues alone. Additional analyses are included in Appendix F.3 and F.4.

**Judge comparison.** We observe that predictions evaluated by ChatGPT-4o and Gemini 2.0 Flash follow consistent overall trends. While Gemini 2.0 Flash reports higher absolute scores in more open-ended tasks such as *Find My Item* and *Action Planning* (around ∼1.5× that of ChatGPT-4o), the relative ordering across models remains largely stable. This indicates that both judges

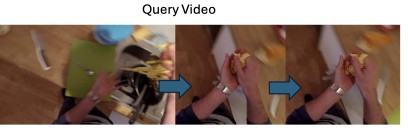
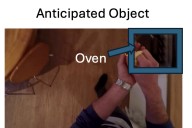

*Figure 5.* Furniture Affordance Prediction example.

agree on the comparative ranking of methods, supporting the robustness of our evaluation.

**Cross-Dataset Evaluation.** Our REA dataset provides effective supervision that significantly boosts the generalization of MLLMs on diverse 3D-related tasks. In particular, we conduct zero-shot evaluation on SQA3D (Ma et al., 2023), a situated QA benchmark designed to assess scene understanding for embodied agents. As shown in Table 3, our STLLM baselines consistently outperform strong existing models, highlighting the effectiveness of spatio-temporal LLM designs. We also observe that models finetuned on REA (marked with †) achieve clear improvements compared to their vanilla counterparts, demonstrating the transferability of our dataset to downstream 3D reasoning tasks. For instance, LLaVA-Video-7B-Qwen (Zhang et al., 2024) improves its average accuracy from **43–45%** to **52–53%**, and LLaVA-OV-Qwen2-7B (Li et al., 2024) improves from **42–44%** to **44–49%**. These gains highlight that REA training substantially enhances cross-dataset generalization, even for strong existing video-language models. As a complementary setting with static multi-image inputs rather than egocentric video and 3D point clouds, we also evaluate zero-shot transfer on MMSI-Bench (Yang et al., 2025c); full results are provided in Appendix F.2.

**Qualitative results.** Fig. 5 presents a query video for a case from the *Furniture Affordance* Prediction task, where the query asks: *Which object will the person interact with next, the oven or the fridge?* The video shows the person completing an action at the sink, and the model must anticipate the next likely interaction. STLLM models correctly predict that *"The person is preparing to interact with*

*the oven, as they are moving closer to it,"* capturing the spatial intention toward a valid future object. In contrast, LLaVA-Video-7B-Qwen2 (Zhang et al., 2024) incorrectly responds that the person is *"preparing to interact with the sink,"* a response grounded in the current frame rather than a forward-looking prediction. This highlights a key limitation of existing models: they often rely on immediate visual context without reasoning about temporal progression. In contrast, as intended spatio-temporal LLMs demonstrate compelling spatio-temporal reasoning.

## 6. Conclusion

Joint spatio-temporal reasoning about an unfolding event given egocentric observations as well as a predicted allocentric environment representation, which permits to ground temporal local motion in a global context, is a crucial task for embodied AI. To study this capability, we collect the "Reasoning about Environments and Actions" (REA) data, consisting of five tasks, via a developed dataset collection pipeline. We show that classic multi-modal language models (MLLMs) struggle to correctly answer spatio-temporal prompts. We also show that results improve significantly when an MLLM is equipped with pointcloud understanding.

**Limitations.** While we present a first step towards joint spatio-temporal reasoning, more work is needed to better understand 1) the type of training data that is most helpful for improving spatio-temporal reasoning of MLLMs; 2) the MLLM components that best extract meaningful information from the data. As a starting point for item 1 we present REA. As a starting point for item 2 we study two complementary baselines that highlight different trade-offs in extracting spatial information for multimodal reasoning.

**Acknowledgements.** Work supported in part by NSF grants 2008387, 2045586, 2106825, MRI 1725729, and NIFA award 2020-67021- 32799.

## Impact Statement

We envision significant positive broader impacts as spatio-temporal reasoning is a crucial component for embodied AI. However, successful spatio-temporal reasoning can also be abused for unnoticed mass surveillance. Hence, deployment of such technology requires great care.

## Acknowledgements

This work is supported in part by NSF grants 2008387, 2045586, 2106825, CNS 2106592, OAC 2126246, and NIFA award 2020-67021-32799. This research used the Delta advanced computing and data resource, which is supported by the National Science Foundation (award OAC 2005572) and the State of Illinois. The opinions expressed in this publication are those of the authors and do not necessarily reflect those of the funding agencies.

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

## Appendix: Spatio-Temporal LLM: Reasoning about Environments and Actions

This appendix is organized as follows: Sec. A provides the definition for the 5 tasks in our dataset. Sec. B discusses the reliability of our dataset and the LLM-Judge metrics. Sec. C provides the LLM Judge prompts used for evaluation. In Sec. D, we include the inference instruction prompts used for baseline video models. Sec. E outlines training details. Sec. F contains additional experiments and analyses. Sec. G showcases additional qualitative results. Sec. H provides further details about the **REA** dataset. Sec. I discusses limitations. Finally, Sec. J discusses the LLM usage in this work.

## A. Task Definition.

In this section, we describe the detailed definition of the 5 tasks:

**Relative Direction.** This task requires to analyze relative direction transitions of an object in the 3D scene during a series of actions performed by the person recording the egocentric video. The question asks to infer how the person's body orientation changes w.r.t. an object across two actions, such as inferring whether the hand closer to an object differs for two consecutive actions. The task can involve a single object or multiple objects. In the **single-object** setting, the question asks whether the person's movement or change in body orientation across a sequence of actions has resulted in a shift in the object's relative direction w.r.t. the person. In the **multi-object** setting, the question asks about the spatial relationship between multiple objects and the person, assessing whether their relative position or the person's viewpoint toward them remains consistent.

**Relative Distance.** This task evaluates the ability to reason about how the person's proximity to one or more objects changes over time, requiring spatial awareness across two query actions in the **single-object** setting and comparative distance understanding in the **multi-object** setting. Specifically, we ask questions such as "*Does the person move closer to the query object between the query action 1 and query action 2?*" (single-object), and "*During the first query action, is the person closer to the query object than to the reference object?*" (multi-object).

**Find My Item.** The task assesses the ability to localize an object and infer spatial steps to reach it, requiring integration of scene understanding and movement planning. An example question is: "*After performing the query action, where did the person leave the query object and how to reach it?*". The model must identify the object's placement from the video and reason about the spatial path an agent would take to reach it.

**Furniture Affordance Prediction.** This task requires to predict which static furniture object the person is likely to interact with next, based on visual observations from the input video and spatial cues about the surrounding environment. The model must infer the recent action sequence, movement trend, and nearby layout. E.g., a question reads as "*Based on what the person has done so far and how they're moving now, which nearby object is the person preparing to interact with?*"

**Action Planning.** This task evaluates the ability to anticipate the next action in a task sequence and provide a navigation instruction to reach the location where the next step will occur. Crucially, the target location is often not directly visible in the egocentric video, requiring the model to reason over the 3D point cloud representation of the environment to localize the current position and plan a path toward the anticipated interaction site.

## B. Dataset and Metrics Reliability

### B.1. Dataset

*Table 4.* Human Evaluation on a Subset of the Test Split (values are accuracy in %).

| Task | PCD Quality | QA Quality | Spatio-Temporal | Spatial Relation | Temporal Logic | Semantic Correctness | Clarity of Question |
|---|---|---|---|---|---|---|---|
| Relative Direction | 80.00 | 80.00 | 80.00 | 85.00 | 100.00 | 95.00 | 100.00 |
| Relative Distance | 90.00 | 80.00 | 95.00 | 80.00 | 100.00 | 100.00 | 100.00 |
| Find My Item | 100.00 | 80.00 | 65.00 | 75.00 | 100.00 | 100.00 | 90.00 |
| Furniture Affordance | 85.71 | 80.95 | 85.71 | 90.48 | 100.00 | 90.48 | 100.00 |
| Action Planning | 100.00 | 85.00 | 75.00 | 95.00 | 85.00 | 90.00 | 100.00 |
| Overall | 91.09 | 81.19 | 80.20 | 85.15 | 97.03 | 95.05 | 98.02 |

We implemented manual quality control during dataset collection to ensure the reliability of both the QA pairs and the VideoLLM-based refinement process. Specifically, we performed human verification and evaluation on the generated VQA pairs in the test set. We conducted the human evaluation on 100 samples (20 per task), where two expert annotators independently reviewed each sample. A score was marked as 1 only if both annotators agreed it was correct. We evaluate

the test set using the following criteria:

- Point Cloud Quality: Does the point cloud accurately capture the positions of the reference objects?

- QA Quality: Scored as 1 only if all of the following five sub-criteria are rated 1: spatial relation, temporal logic, semantic correctness, and clarity of the question; otherwise, 0.

- Spatio-Temporal Reasoning: Does the question require understanding of both spatial layout and temporal action sequence? Does the answer demonstrate such reasoning?

- Spatial Relation: Is the spatial relationship between the query object and the person accurately described in the answer?

- Temporal Logic: Does the video contain the actions mentioned in the question in a temporally coherent manner?

- Semantic Correctness: Does the answer correctly and clearly explain the reason behind the movement or action?

- Clarity of Question: Is the question phrased clearly, fluently, and naturally, as if written by a human?

The REA dataset is specifically designed so that the query video often does not directly show the query objects, making point cloud information essential for spatial reasoning. As shown in Table 4, to answer correctly, 80% of the questions require access to the point cloud.

### B.2. LLM-Judge

We have incorporated human evaluation to refine the LLM Judge prompts. Specifically, after each prompt update, we randomly sampled 40 examples per task and compared the LLM Judge's verdicts with human judgments. We iteratively refined the prompt until the agreement between the LLM Judge and human evaluation exceeded 97%. Please refer to Appendix C for the final LLM Judge prompt. We note that the LLM Judge is particularly strict on tasks such as *Find My Item* and *Action Planning* as these involve open-ended question answering. This strictness helps ensure consistent and high-quality evaluation for tasks with less constrained answers. Also note that the LLM judge is asked to compare a given answer predicted by the models we study to the given ground truth answer. Hence, the LLM judge is not required to solve the spatio-temporal reasoning task. Instead, the LLM judge is tasked to assess the equivalence of the provided statements.

## C. LLM Judge Prompt

We design task-specific prompts for the LLM Judge. The prompt is the same for `ChatGPT-4o` and `Gemini 2.0 Flash`.

---

**Relative Direction**

You are a helpful and fair evaluator. Your task is to determine whether the predicted answer correctly follows the ground truth answer for a relative direction query. This task involves reasoning about the directional relationship between the person and a referenced object during two different actions, based on the scene.
**Predicted Result**: {pred}
**Ground Truth Result**: {gt}
**Query**: {query}
Please answer only with "Correct" or "Wrong", based on the following criteria:
- Mark as "Correct" if the predicted answer accurately reflects the relative direction of the object (e.g., left, right, forward, behind) in relation to the person across both mentioned actions, even if the wording differs.
- Directional terms may vary slightly (e.g., "in front" vs. "forward") but must preserve spatial meaning.
- The answer must address both actions in the query.
- If the prediction misidentifies one or both relative directions, or skips one, mark it as "Wrong".
Only reply with one word: "correct" or "wrong" — no explanation or extra text. If the prediction matches the ground truth, reply "correct". Otherwise, reply "wrong".

---

**Relative Distance**

You are a helpful and fair evaluator. Your task is to determine whether the predicted answer correctly follows the ground truth answer for a relative distance query. This task involves comparing the person's distance to a specific object during two different actions, based on the scene.
**Predicted Result**: {pred}
**Ground Truth Result**: {gt}
**Query**: {query}
Please answer only with "Correct" or "Wrong", based on the following criteria:
- Mark as "Correct" if the predicted answer accurately conveys the relative distance relationship described in the ground truth, even if expressed with different wording.
- The prediction must clearly indicate which action places the person closer (or if the distances are about the same).
- Minor wording variations or additional clarifications are acceptable as long as the core spatial relationship is preserved.
- If the prediction contradicts or misses the comparison stated in the ground truth, mark it as "Wrong".
- Move closer and move further sometimes can be similar to remain the same distance, based on the context of the prediction, give reasonable judgement.
Only reply with one word: "correct" or "wrong" — no explanation or extra text. If the prediction matches the ground truth, reply "correct". Otherwise, reply "wrong".

**Find My Item**

You are a helpful and fair evaluator. Your task is to determine whether the predicted answer correctly follows the ground truth answer for a 'Find My Item' query. This task requires identifying the location of a target object and describing how the person can get to it, based on the scene.
**Predicted Result**: {pred}
**Ground Truth Result**: {gt}
**Query**: {query}
Please answer only with "Correct" or "Wrong", based on the following criteria:
- Mark as "Correct" if the predicted answer matches the essential intent and meaning of the ground truth, even if phrased differently.
- The answer must correctly identify the item's location and provide a reasonable description of how to reach it.
- Minor differences in language, additional helpful navigation details, or alternative phrasing are acceptable if the overall meaning is consistent with the ground truth.
- If the predicted answer omits key information, misidentifies the item's location, or gives an implausible or unrelated navigation instruction, mark it as "Wrong".
Only reply with one word: "correct" or "wrong" — no explanation or extra text. If the prediction matches the ground truth, reply "correct". Otherwise, reply "wrong".

**Furniture Affordance**

You are a helpful and fair evaluator. Your task is to determine whether the predicted answer correctly follows the ground truth answer for a furniture affordance query. This task involves reasoning about the person's past actions and current movement to infer which nearby object they are most likely preparing to interact with.

**Predicted Result**: {pred}

**Ground Truth Result**: {gt}

**Query**: {query}

Please answer only with "Correct" or "Wrong", based on the following criteria:

- Mark as "Correct" if the prediction correctly identifies the most likely object of interaction based on the query and provides a valid rationale aligned with the ground truth.

- The predicted object must match the correct option (e.g., "oven" or "fridge").

- Minor differences in phrasing or additional reasoning are acceptable as long as the predicted object is the same and the rationale is plausible.

- If the prediction identifies the wrong object or gives an unreasonable explanation, mark it as "Wrong".

Only reply with one word: "correct" or "wrong" — no explanation or extra text. If the prediction matches the ground truth, reply "correct". Otherwise, reply "wrong".

**Action Planning**

You are a helpful and fair evaluator. Your task is to determine whether the predicted answer correctly follows the ground truth answer for an action planning query. The action planning task involves reasoning about sequences of actions in a cooking or assembly video, and determining what to do next and how to get there.

**Predicted Result**: {pred}

**Ground Truth Result**: {gt}

**Query**: {query}

Please answer only with "Correct" or "Wrong", based on the following criteria:

- Mark as Correct if the predicted answer matches the intent and content of the ground truth, even if the wording is different. Reasonable paraphrasing is acceptable.

- The answer must identify the correct next step in the sequence, based on the context.

- It must also provide a plausible description of how to reach the location of the next step.

- Minor differences in phrasing or additional helpful details are acceptable, as long as the core actions are logically consistent with the ground truth.

- Avoid over-penalizing answers for surface-level differences if they preserve the meaning and ordering of actions.

Only reply with one word: "correct" or "wrong" — no explanation or extra text. If the prediction matches the ground truth, reply "correct". Otherwise, reply "wrong".

> **SQA3D**
>
> You are a helpful and fair evaluator. Your task is to determine whether the predicted answer correctly follows the ground truth answer for a furniture affordance query. This task involves reasoning about the person's past actions and current movement to infer which nearby object they are most likely preparing to interact with.
> **Predicted Result**: {pred}
> **Ground Truth Result**: {gt}
> **Query**: {query}
> Please answer only with "Correct" or "Wrong", based on the following criteria:
> - Mark as "Correct" if the predicted answer expresses or implies the correct object or direction mentioned in the ground truth, even if phrased as a sentence, includes assistant prefixes, or contains extra context.
> - Slight mismatches, rephrasings, or formatting issues (e.g., "The suitcase is under the bed." vs. "suitcase") are acceptable as long as the prediction clearly reflects the correct meaning.
> - Mark as "Wrong" only if the answer refers to an entirely different object, contradicts the spatial context, or fails to address the question meaningfully.
> Only reply with one word: "correct" or "wrong" — no explanation or extra text. If the prediction matches the ground truth, reply "correct". Otherwise, reply "wrong".

## D. Inference Instruction Prompts for VideoLLMs

As the existing VideoLLMs cannot take the point cloud as input, we use the 25 multi-view images (used for point cloud reconstruction) as a static scene description, and then input the same 32 query video frames as in our models.

To enable fair evaluation by an LLM judge, we prompt these models to generate full-sentence explanations rather than short answers like "yes" or "no". This ensures that the judge assesses answers based on reasoning rather than matching surface-level correctness.

Additionally, since these existing models were not trained to interpret the input as two separate streams (i.e., a static scene and a dynamic query video), we explicitly include this structure in the prompt to guide their attention accordingly.

> **Instruction Prompts**
>
> The first 25 images provide multi-view observations of the current scene the person is in, while the next 32 frames depict egocentric actions—please refer to both to answer the question.
> **<Image>** {question}
> Give explanations and reasoning for your answer. Answer in detail, and be specific. Do not random guess. If you don't know, say 'I don't know'.

# E. Training Details

Our training follows the standard next-token prediction objective, optimizing the token-wise cross-entropy loss over the LLM outputs. During training, modality-specific encoders remain frozen. We finetune the modality projectors and the LLM decoder (adapted with LoRA (Hu et al., 2022)), with STLLM-Aligner additionally updating the alignment module (see Fig. 4).

All training was conducted using four NVIDIA H200 GPUs. The models were trained on the REA dataset for one epoch, which took approximately 6 hours. We adopt single-epoch training to balance efficiency and generalization. The LLM decoder is adapted using LoRA (Hu et al., 2022) finetuning, enabling efficient parameter updates while mitigating overfitting to fixed answer templates and preserving the ability to generalize beyond rigid output structures.

For both models, the point cloud encoder is executed in `float32` precision, whereas the remaining components are trained in `bfloat16` for efficiency. `Nquery` stands for the number of learnable embeddings in the alignment module.

*Table 5.* Training hyperparameters for STLLM-Aligner and STLLM-Aligner (w Pos. Enc.) model.

| Parameter | Value |
|---|---|
| Gradient Accumulation Steps | 8 |
| Learning Rate | $1 \times 10^{-4}$ |
| Weight Decay | 0 |
| Precision$^*$ | bfloat16 |
| Max Frames | 32 |
| Voxel Size | 0.06 |
| Nquery | 1024 |

In Tab. 5, `Nquery` denotes the number of tokens in the learnable query.

*Table 6.* Training hyperparameters for STLLM-3D.

| Parameter | Value |
|---|---|
| Gradient Accumulation Steps | 8 |
| Learning Rate | $1 \times 10^{-4}$ |
| Weight Decay | 0 |
| Precision$^*$ | bfloat16 |
| Max Frames | 32 |
| Voxel Size | 0.06 |
| Npoint | 1024 |
| Radius | 0.2 |
| Nsample | 64 |

In Tab. 6, `Npoint` denotes the number of center points sampled from the point cloud features. For each center point, a local neighborhood is defined by a specified `Radius`, and up to `Nsample` points are gathered within this radius to form a group.

## E.1. System Prompt

We explicitly inform the model that the first `Nquery` visual tokens encode global spatio-temporal context, which it should pay special attention to during reasoning.

> **Instruction Prompts for REA**
>
> The first 1024 tokens encode learnable queries representing objects and locations in the 3D scene. The following tokens represent egocentric video of recent actions. Use both to reason about spatial references and temporal context when answering.

# F. Additional Experiments

*Table 7.* LLM-Judge accuracy (%, higher is better, C = `ChatGPT-4o`, G = `Gemini 2.0 Flash`).

| Model | | Rel. Dir. | Rel. Dist. | Find My Item | Affordance | Action Plan. | Overall / Avg. |
|---|---|---|---|---|---|---|---|
| LLaVA-Video-7B-Qwen2 (Zhang et al., 2024) | C | 36.67 | 43.00 | 28.06 | 53.05 | 13.17 | 30.96 / 34.79 |
| | G | 46.00 | 42.67 | 38.49 | 56.27 | 27.33 | 39.50 / 42.15 |
| LLaVA-OV-Qwen2-7B (Li et al., 2024) | C | 15.33 | 36.00 | 25.54 | 50.18 | 9.00 | 23.85 / 27.21 |
| | G | 36.67 | 40.00 | 40.65 | 51.61 | 23.50 | 35.74 / 38.49 |
| Qwen2-VL-7B-Instruct (Wang et al., 2024a) | C | 38.33 | 9.67 | 15.47 | 40.50 | 15.00 | 24.38 / 23.68 |
| | G | 36.67 | 10.00 | 23.02 | 41.22 | 33.67 | 29.94 / 27.90 |
| VideoLLaMa3 (Zhang et al., 2025a) | C | **57.00** | 42.00 | 20.86 | 39.43 | 10.00 | 31.46 / 35.86 |
| | G | **70.33** | 38.33 | 42.45 | 39.07 | 13.33 | 36.03 / 40.70 |
| Qwen2.5-VL-7B (Bai et al., 2025) | C | 33.00 | 40.67 | 16.91 | 49.82 | 14.17 | 28.00 / 30.91 |
| | G | 48.67 | 34.33 | 29.14 | 50.54 | 21.83 | 34.26 / 36.90 |
| Qwen3-VL-8B (Team, 2025) | C | 29.67 | 32.33 | 14.39 | 26.88 | 16.67 | 22.82 / 23.99 |
| | G | 40.00 | 31.33 | 34.89 | 50.18 | **47.83** | 42.00 / 40.85 |
| InternVL3.5-8B (Wang et al., 2025b) | C | 28.00 | 47.33 | 28.06 | 28.67 | 15.00 | 26.98 / 29.41 |
| | G | 45.00 | 49.33 | 44.60 | 35.13 | 23.83 | 36.88 / 39.58 |
| *Proprietary API model* | | | | | | | |
| GPT-5.2 | C | 40.33 | 48.00 | 34.89 | 58.06 | 21.17 | 37.05 / 40.49 |
| | G | 33.33 | 42.00 | 43.88 | 58.42 | 31.33 | 39.78 / 41.79 |
| LLaVA-Video-7B-Qwen2[†] | C | 40.67 | 61.00 | 36.69 | 61.65 | 11.83 | 36.99 / 42.37 |
| | G | 44.00 | 61.00 | 56.12 | 57.35 | 20.50 | 42.92 / 47.79 |
| LLaVA-Video-7B-Qwen2[†‡](25 spatial tokens) | C | 29.33 | 53.67 | 29.50 | 61.29 | 8.33 | 31.42 / 36.42 |
| | G | 31.86 | 53.67 | 46.04 | 56.63 | 19.33 | 37.48 / 41.52 |
| LLaVA-OV-Qwen-7B[†] | C | 41.00 | 66.00 | 32.73 | 59.86 | 15.33 | 38.19 / 42.98 |
| | G | 47.00 | 66.00 | 47.12 | 55.91 | 23.50 | 43.65 / 47.91 |
| LLaVA-OV-Qwen2-7B[†‡](25 spatial tokens) | C | 33.00 | 47.00 | 33.09 | 54.84 | 10.17 | 31.08 / 35.62 |
| | G | 35.67 | 47.33 | 49.64 | 48.75 | 20.00 | 36.60 / 40.28 |
| STLLM-3D[‡] (32 Nquery) | C | 44.67 | 32.67 | 26.62 | 63.44 | 12.17 | 31.65 / 35.91 |
| | G | 46.33 | 39.67 | 40.29 | **62.37** | 22.33 | 38.59 / 42.40 |
| STLLM-Aligner[‡] (w Pos. Enc., 32 Nquery) | C | 40.33 | 46.00 | 34.89 | 60.22 | 13.83 | 34.55 / 39.05 |
| | G | 45.00 | 47.67 | 48.20 | 60.22 | 19.00 | 39.50 / 44.02 |
| STLLM-Aligner[‡] (w Pos. Enc.) | C | 56.67 | 49.67 | 28.78 | 63.80 | 7.50 | 35.38 / 41.26 |
| | G | 39.67 | 48.33 | 48.20 | 55.91 | 14.17 | 36.37 / 41.26 |
| STLLM-Aligner (w Pos.Enc.) | C | 49.00 | 69.00 | **38.13** | 59.50 | 17.00 | 41.43 / 46.53 |
| | G | 50.00 | 69.00 | 55.40 | 53.41 | 24.33 | 45.87 / 50.43 |
| **STLLM-Aligner** | C | 50.67 | **70.67** | 36.69 | 62.72 | 15.83 | **41.89 / 47.32** |
| | G | 51.33 | **70.67** | 55.04 | 55.56 | 23.83 | **46.50** / 51.29 |
| **STLLM-3D** | C | 48.00 | 68.00 | 35.61 | **65.69** | 14.83 | 40.94 / 46.43 |
| | G | 51.00 | 68.00 | 56.47 | 58.06 | 23.17 | 46.39 / **51.34** |

*Note.* [†] indicates the existing model is finetuned on our REA dataset. [‡] indicates LLM layers are not finetuned.

## F.1. Quantitative Results

**Finetuned Existing Models.** We pool the multi-view images in LLaVA-Video-7B-Qwen2[†] and LLaVA-OV-Qwen-7B[†] into token sequences similar in length to our model's spatial queries (36 tokens per image) for a fair comparison.

**Additional Experiments.** We conduct additional experiments to explore two factors: 1) the number of learnable spatial queries (`Nquery`), and 2) the number of trainable parameters. For a fair comparison with a smaller number of spatial queries (32 `Nquery`), we also finetune strong baseline models, namely LLaVA-Video-7B-Qwen[†‡](25 spatial tokens) and LLaVA-OV-Qwen-7B[†‡](25 spatial tokens), both of which represent spatial information using one token per multi-view image (25 multi-view images in total). As shown in Table 7, STLLM-Aligner[‡] (w/ Pos. Enc.), which uses 1024 `Nquery`, outperforms STLLM-Aligner[‡] (w/ Pos. Enc., 32 `Nquery`), demonstrating that a larger number of spatial queries leads to better performance. However, the improvement is relatively modest. This motivates us to increase the number of trainable parameters. By further adapting the LLM layers with LoRA (Hu et al., 2022), performance improves substantially: overall

*Table 8.* Comparison of models on various evaluation metrics. Sim = Sentence Similarity.

| Model | Sim (%) ↑ | CIDEr ↑ | BLEU (%) ↑ | METEOR (%) ↑ | ROUGE (%) ↑ |
|---|---|---|---|---|---|
| LLaVA-Video-7B-Qwen2 (Zhang et al., 2024) | 65.83 | 20.79 | 10.25 | 19.68 | 23.71 |
| LLaVA-OV-Qwen2-7B (Li et al., 2024) | 64.51 | 3.34 | 11.22 | 19.53 | 23.84 |
| Qwen2-VL-7B-Instruct (Wang et al., 2024a) | 52.99 | 35.08 | 19.11 | 17.38 | 25.37 |
| VideoLLaMa3 (Zhang et al., 2025a) | 39.14 | 10.85 | 2.15 | 7.85 | 14.12 |
| LLaVA-Video-7B-Qwen2[†‡](25 spatial tokens) | 81.76 | 304.05 | 48.63 | 34.45 | 59.55 |
| LLaVA-Video-7B-Qwen2[†] | 85.26 | 387.72 | 60.34 | 43.18 | **72.09** |
| LLaVA-OneVision-Qwen-7B[†‡](25 spatial tokens) | 81.06 | 297.79 | 45.43 | 33.94 | 58.45 |
| LLaVA-OneVision-Qwen-7B[†] | 85.09 | 400.23 | 61.90 | 42.41 | 71.11 |
| STLLM-3D[‡] (32 Nquery) | 73.27 | 141.29 | 37.65 | 28.87 | 48.78 |
| STLLM-Aligner[‡] (w Pos.Enc.) | 71.34 | 170.63 | 39.46 | 28.54 | 50.02 |
| STLLM-Aligner (w Pos. Enc.) | 85.70 | 406.54 | 61.90 | **44.16** | **72.09** |
| **STLLM-Aligner** | 85.58 | **406.68** | **62.01** | 43.94 | 72.03 |
| **STLLM-3D** | **85.99** | 405.48 | 61.99 | 44.04 | 72.07 |

accuracy rises from 35.38% / 36.37% (C/G) for STLLM-Aligner[‡] (w/ Pos. Enc.) to 41.43% / 45.87% for STLLM-Aligner (w/ Pos. Enc.). Under all settings, as shown in both Table 7 and Table 8, our STLLM baselines outperform the direct finetuned counterparts, which showcase the effectiveness of our model designs.

**Strong proprietary baseline.** We further evaluate GPT-5.2 as a representative proprietary API model on REA. As shown in Table 7, GPT-5.2 (OpenAI, 2025) outperforms recent open-source baselines under the ChatGPT-4o judge, achieving 37.05% overall accuracy compared to the best open-source baseline at 31.46%. However, it still remains below our fine-tuned STLLM baselines, which achieve 41.89% for STLLM-Aligner and 40.94% for STLLM-3D. A similar trend holds under the Gemini judge, where GPT-5.2 obtains 39.78% overall accuracy, while STLLM-Aligner and STLLM-3D reach 46.50% and 46.39%, respectively. These results further show that REA remains challenging even for strong proprietary MLLMs, and that explicit spatio-temporal modeling provides consistent gains.

*Table 9.* Zero-shot cross-dataset evaluation on MMSI-Bench.

| Model | Positional Relationships | | | | | | Attributes | | Motion | | MSR | Avg. |
|---|---|---|---|---|---|---|---|---|---|---|---|---|
| | Cam.-Cam. | Obj.-Obj. | Reg.-Reg. | Cam.-Obj. | Obj.-Reg. | Cam.-Reg. | Meas. | Appr. | Cam. | Obj. | | |
| LLaVA-Video-7B-Qwen2 (Zhang et al., 2024) | 28.0 | **30.9** | 23.5 | **36.0** | **40.0** | **33.7** | 25.0 | **34.8** | 14.9 | 28.9 | **23.7** | **28.6** |
| STLLM-Aligner | **36.6** | 27.7 | **25.9** | 33.7 | 32.9 | 32.5 | **26.6** | 27.3 | 12.2 | **28.9** | 20.7 | 27.2 |

*Table 10.* Video-only ablation under ChatGPT-4o judge accuracy (%).

| Model | Rel. Dir. | Rel. Dist. | Find My Item | Furn. Aff. | Act. Plan. | Overall |
|---|---|---|---|---|---|---|
| LLaVA-Video-7B-Qwen2 (video only) | 38.3 | 43.3 | 18.7 | 52.0 | 10.0 | 28.6 |
| STLLM-Aligner (video only) | 43.3 | 55.0 | 30.9 | 65.2 | 13.3 | 36.6 |
| **STLLM-Aligner** | 50.7 | 70.7 | 36.7 | 62.7 | 15.8 | 41.9 |
| **STLLM-3D** | 48.0 | 68.0 | 35.6 | 65.7 | 14.8 | 40.9 |

*Table 11.* Removing the point-cloud branch leads to a clear performance drop, especially on spatially demanding tasks.

## F.2. Additional Cross-Dataset Evaluation

As a complementary cross-dataset analysis, we further evaluate zero-shot transfer on MMSI-Bench (Yang et al., 2025c), a static multi-image spatial reasoning benchmark. As shown in Table 9, MMSI-Bench differs from REA in that it does not provide egocentric video or 3D point-cloud inputs, creating a paradigm mismatch with our training setting. Under this setting, STLLM-Aligner obtains comparable overall performance to the LLaVA-Video-7B-Qwen2 base model (**27.2%** vs. **28.6%**). While appearance-oriented categories drop, STLLM-Aligner improves on several spatial relation categories, including camera-camera relations (**36.6%** vs. **28.0%**) and region-region relations (**25.9%** vs. **23.5%**). This suggests that REA training transfers most clearly to geometric viewpoint reasoning, while benefits are less pronounced when the target benchmark removes the allocentric 3D context used by our STLLM models.

## F.3. Ablation

To assess the contribution of allocentric 3D context, we further evaluate a video-only variant by removing the point-cloud branch from our current setting. Removing the point cloud reduces ChatGPT-4o judge accuracy from **41.9%** to **36.6%**, with the largest drops on spatially demanding tasks such as Relative Distance (**70.7%** to **55.0%**) and Relative Direction (**50.7%** to **43.3%**). This confirms that the point-cloud branch contributes complementary allocentric spatial information that cannot be recovered from egocentric video alone.

## F.4. Attention Analysis

We visualized the LLM decoder attention patterns for both baselines to examine whether the models truly perform spatio-temporal reasoning. We observe three clear patterns from Fig. 6:

**Global spatial anchors.** Point-cloud tokens form a stable, narrow attention band in the baselines, indicating that the decoder repeatedly references the global 3D layout as spatial anchors. The tilted pattern arises because different decoder query positions attend to different groups of point-cloud tokens, reflecting that the models actively use structured 3D features rather than collapsing onto a fixed set of spatial tokens.

**Local visual detail extraction.** Video tokens show high-frequency, fine-grained attention textures, demonstrating that the model actively attends to local visual details (specific object or hand-object regions) rather than treating frames as global features. Moreover, as shown in Fig. 6, the decoder places stronger attention on the image tokens near the beginning and ending frames of the video, which correspond precisely to where the queried actions occur, indicating correct temporal grounding.

**Strong Instruction Grounding.** Text tokens receive consistently stronger attention to the system prompt, showing that predictions remain closely tied to the query instructions.

Taken together, the attention patterns suggest that the model incorporates all textual, temporal, and 3D spatial cues during reasoning, and that the 3D spatial cues further contribute to the performance gains. We include an ablation supporting this:

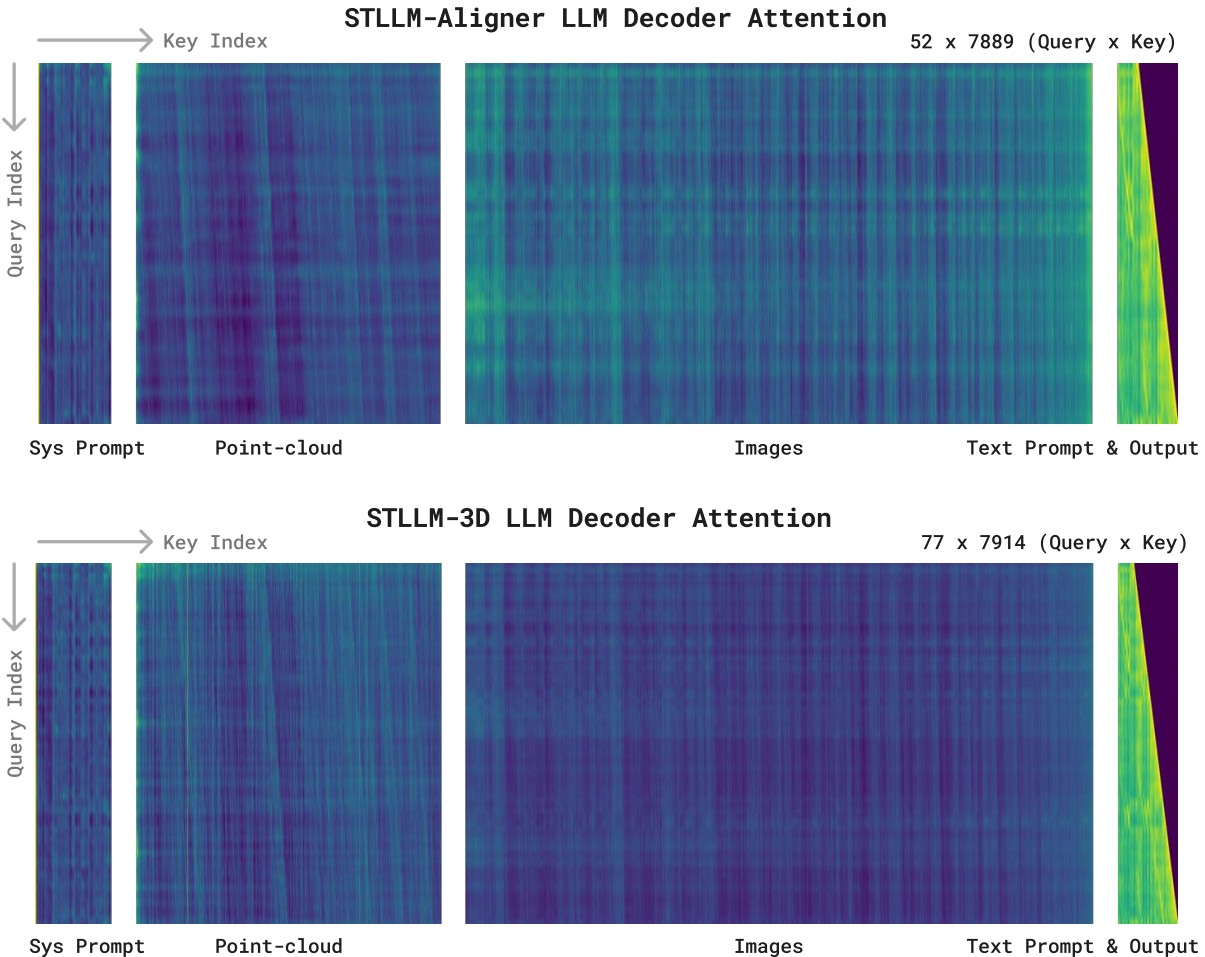

*Figure 6.* LLM decoder attention visualization of STLLM baselines. Query Index represents the positions of decoder-side queries, and Key Index represents the positions of all tokens available to the decoder (system prompt, point-cloud, images, and previously generated output tokens).

replacing the 3D point-cloud with a pure 2D multi-view input causes a clear drop in accuracy (41.89 to 36.99 overall; 47.32 to 42.37 average in Table 2), indicating that the model benefits from reasoning over explicit 3D geometry rather than relying on frame-level appearance cues alone.

# G. Qualitative Results

In the qualitative results below, ✓ and ✗ indicate whether the prediction was marked as correct or incorrect by `ChatGPT-4o`, which serves as our LLM judge. Each figure shows two representative frames sampled from the query video, rendered below the reconstructed point cloud of the scene to provide spatial context for the queried actions.

---

**Relative Direction**

**Question**: Is the microwave to the left of the person when the person is performing 'put down knives' and also to the left of the person when performing 'pick up plate'?
**Ground-truth**: The microwave remains to the left of the person during both 'put down knives' and 'pick up plate'.
**Ours**: The microwave is to the left of the person before and after both 'put down knives' and 'pick up plate'. ✓
**LLaVA-OV-Qwen2-7B**: The microwave is not to the left of the person when they perform "put down knives" and "pick up plate." The sequence shows that the person stands in front of the counter where the microwave is located, and they are not interacting with it during those specific actions. ✗

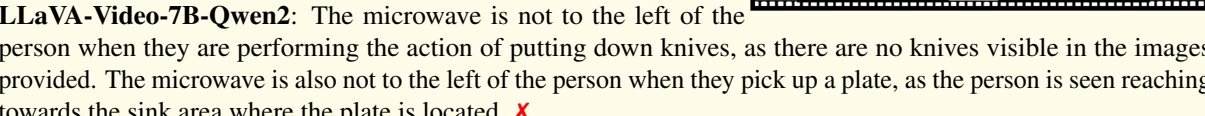

**LLaVA-Video-7B-Qwen2**: The microwave is not to the left of the person when they are performing the action of putting down knives, as there are no knives visible in the images provided. The microwave is also not to the left of the person when they pick up a plate, as the person is seen reaching towards the sink area where the plate is located. ✗
**Qwen2-VL-7B-Instruct**: No, the microwave is not to the left of the person when the person is performing 'put down knives' and to the left of the person when 'pick up plate'. ✗

---

**Relative Distance**

**Question**: Does the person move closer to the hob between 'turn off rice cooker' and 'serve curry'?
**Ground-truth**: The person remains at about the same distance from the hob when performing both 'turn off rice cooker' and 'serve curry'.
**Ours**: The person remains at about the same distance from the hob when performing both 'turn off rice cooker' and 'serve curry'. ✓
**LLaVA-OV-Qwen2-7B**: The person does not move closer to the hob between "turn off rice cooker" and "serve curry". The sequence of actions shows that after turning off the rice cooker, the person moves towards the sink area. The focus then shifts back to the countertop where the rice and curry are being served, but there is no direct interaction with the hob during these transitions. ✗

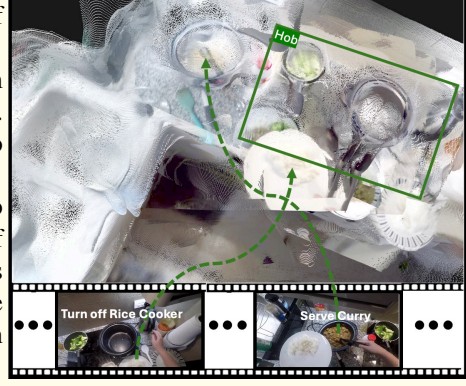

**LLaVA-Video-7B-Qwen2**: Yes, the person moves closer to the hob between 'turn off rice cooker' and 'serve curry'. The sequence of actions suggests that after turning off the rice cooker, the person is likely to be preparing to serve the dish. Moving closer to the hob implies that they are in the process of transferring the cooked curry onto the rice plates. ✗
**Qwen2-VL-7B-Instruct**: Yes, the person moves closer to the hob between 'turn off rice cooker' and 'serve curry'. In the first image, the person is standing further away from the hob, while in the second image, the person is standing closer to the hob. This can be inferred from the relative positions of the objects in the images. ✗

## Find My Item

**Question**: Where is the tarragon, and how can the person get to it?
**Ground-truth**: The tarragon is located on the countertop to the right of the person. The person can reach it by turning to their right and walking towards the countertop.
**Ours**: The tarragon is on the countertop to the right of the person. The person can reach it by turning to their right and walking towards the countertop. ✓
**LLaVA-OV-Qwen2-7B**: The tarragon appears to be on the countertop, specifically on a plate or tray with other chopped vegetables. To get to it, the person would need to move around the kitchen counter and reach towards the area where the tarragon is placed. ✗
**LLaVA-Video-7B-Qwen2**: The tarragon is on the counter in front of the person. The person can get to it by moving closer to the counter and picking up the bowl containing the tarragon. ✗
**Qwen2-VL-7B-Instruct**: The tarragon is located on the kitchen counter. The person can get to it by walking towards the counter and reaching for the tarragon. ✓

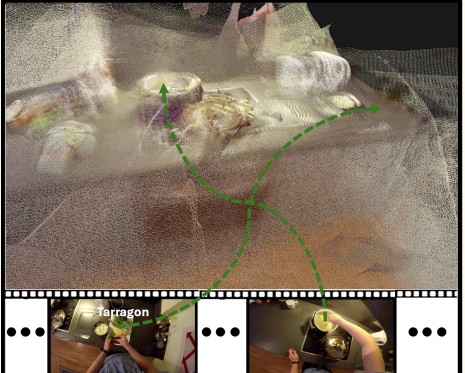

## Furniture Affordance

**Question**: Which of the following objects does the person interact with next, given their previous actions and current motion?
A. hob, B. oven, C. fridge
**Ground-truth**: The person is most likely to interact with the hob next because they have just finished cleaning the sink and are now moving towards the hob, which suggests they might need to use it for cooking or cleaning purposes.
**Ours**: The person is most likely to interact with the hob next because they have just finished washing dishes and are now moving towards the hob, which is typically used for cooking. ✓
**LLaVA-OV-Qwen2-7B**: B. oven. ✗
**LLaVA-Video-7B-Qwen2**: B. oven ✗
**Qwen2-VL-7B-Instruct**: I don't know. ✗

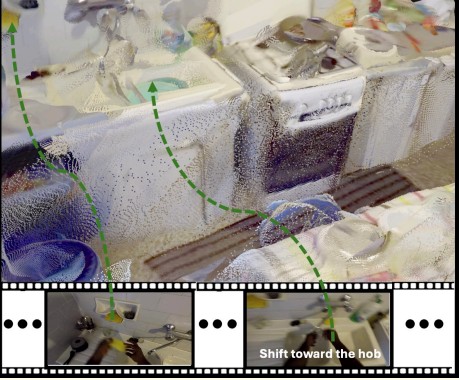

**Action Planning**

**Question**: We are performing a cooking/assembly task with the following sequence of actions: put down tray, put down sponge, take glass, wash knife.. Based on the video, what should I do next, and how can I get to the place where the next step takes place?

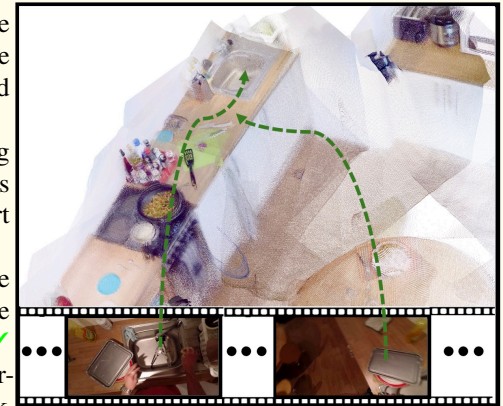

**Ground-truth**: You have already completed the actions of putting down the tray and the sponge. Now, you are about to take the glass and wash the knife. To do this, move right to the sink and start washing the knife.

**Ours**: You have already completed the actions of putting down the tray and the sponge, and taking the glass. Now, you need to wash the knife. To do this, move right to the sink and wash the knife there. ✓

**LLaVA-OV-Qwen2-7B**: Based on the video, it appears that the person is in the process of cleaning up and preparing for another task. The sequence of actions you've mentioned suggests that the person has already put down a tray and a sponge, taken a glass, and washed a knife. The next step would likely involve using the washed knife to prepare food or assemble something. To determine the next step and where it takes place, let's analyze the last few frames: 1. The person is standing near the sink area. 2. There is a wooden table with a glass and a can nearby. 3. A hand wearing a watch is visible, which might belong to the person performing these tasks. Considering these details, the next logical action could be to use the knife to cut or chop ingredients, which would typically be done on a cutting board. Since the person is already near the sink, they might move towards the wooden table where there is a glass and a can, possibly to find additional ingredients or tools needed for the task. Given the context, the next step would likely involve placing the knife on the table, perhaps on a cutting board if one is available, and then preparing food or assembling an item using the knife. If a cutting board is not present, the person might need to locate one. ✗

**LLaVA-Video-7B-Qwen2**: The next step in the cooking/assembly task should be to put the washed knife back into the sink or a designated storage area. To get to the place where this step takes place, the person should move towards the sink area where the knife was washed earlier. The sink is located on the left side of the frame, adjacent to the countertop with the red plate and the pizza. The person can then reach into the sink to place the knife back, ensuring it is properly cleaned and stored. ✗

**Qwen2-VL-7B-Instruct**: Based on the sequence of actions provided, the next step after washing the knife is to put down the sponge. The reasoning is that after washing the knife, it is typically appropriate to clean the hands or other utensils used, which would involve using the sponge. To get to the place where the next step takes place, you should head towards the sink area. In the video, the person is seen washing the knife in the sink, and then putting down the sponge. Therefore, you should move towards the sink to continue with the cleaning process. ✗

### G.1. LLM Judge Failure Case

We use `ChatGPT-4o` and `Gemini 2.0 Flash` as automatic judges to evaluate model predictions. However, in some cases, the judges produce incorrect judgments due to rigid comparisons. The example below is marked as incorrect by `ChatGPT-4o`, even though the prediction and ground truth both correctly identify the fridge as the next object of interaction and provide reasonable justifications.

The only difference lies in the explanation detail: the ground truth states the person placed cheese into the fridge, while the prediction assumes they took cheese and are moving toward the fridge. This type of reasoning difference is speculative and acceptable, as the core intent and predicted outcome remain the same.

---

**Furniture Affordance**

**Question**: Which of the following objects does the person interact with next, given their previous actions and current motion?
A. fridge, B. oven

**Ground-truth**: The person is most likely to interact with the fridge next because they have just placed the mozzarella inside it, and their current movement is towards the fridge, indicating they are likely to retrieve or interact with something else from it.

**Ours**: The person is most likely to interact with the fridge next because they have just taken a piece of cheese and are now moving towards the fridge, which is a common place to store cheese. ✗

---

### G.2. Model Failure Prediction

**Relative Direction**

**Question**: Are the hob and freezer on the same side of the person when performing 'open drawer'?

**Ground-truth**: No, the hob is in front of the person, while the freezer is to the left of the person when the person performing 'open drawer'.

**Ours**: Yes, both the hob and freezer are to the left of the person when performing 'open drawer'. ✗

# H. Dataset Details

## H.1. Dataset Statistics

We provide detailed statistics of our REA dataset. Table 12 reports dataset statistics by tasks, showing the number of question-answer pairs for each task across training and validation splits. Table 13 presents the dataset overview, including video duration, QA counts, average video length, and the number of unique objects and actions.

*Table 12.* REA dataset statistics by tasks.

| Task | Train | Validation |
|------|-------|------------|
| Relative Distance | 4,796 | 300 |
| Relative Direction | 4,765 | 300 |
| Furniture Affordance | 4,192 | 279 |
| Action Planning | 6,500 | 600 |
| Find My Item | 4,118 | 278 |
| Total | 24,371 | 1,757 |

*Table 13.* Dataset Overview.

| Split | Video Duration (hrs) | # Video IDs* | # QAs | Avg Video Duration (sec) | # Unique Objects | # Unique Actions |
|-------|----------------------|--------------|-------|--------------------------|------------------|-------------------|
| Train | 221.80 | 152 | 24,371 | 32.76 | 299 | 4,759 |
| Test | 23.16 | 71 | 1,757 | 47.46 | 94 | 1,309 |

\* We refer to the `video_id` in EPIC-KITCHENS (Damen et al., 2018).

## H.2. Dataset Comparison

**Spatial Understanding Datasets.** Table 14 compares our REA dataset with several well-known related datasets. We include Nymeria (Ma et al., 2024) and HD-EPIC (Perrett et al., 2025), as well as two additional spatial reasoning benchmarks, VSI-Bench (Yang et al., 2025a) and All-Angles Bench (Yeh et al., 2025). This comparison provides context for the scale and unique properties of our REA dataset, which emphasizes joint spatio-temporal reasoning grounded in both 3D scene structure and egocentric action video.

*Table 14.* Comparison of Spatial Understanding Datasets.

| Dataset | Total Video Duration (hrs) | QA Pairs | RGB Video | 3D Modality | Camera Pose | Labelled 3D Environment |
|---------|----------------------------|----------|-----------|-------------|-------------|-------------------------|
| REA (Ours) | 238.33 | 26.2K | ✓ | Point Cloud + Multi-view Images | ✓ | ✓ |
| HD-EPIC (Perrett et al., 2025) | 41.3 | 26.5K | ✓ | 3D Mesh | ✓ | ✓ |
| Nymeria (Ma et al., 2024) | 300 | 310.5K | ✓ | 3D Point Cloud + Sensors | ✓ | ✓ |
| VSI-Bench (Yang et al., 2025a) | N/A | 5K | ✓ | × | ✓ | ✓ |
| All-Angles Bench (Yeh et al., 2025) | N/A | 2.1K | × | Multi-view Images | ✓ | × |

**Embodied Agent Datasets.** REA is relevant to embodied agent research, as it evaluates spatial reasoning capabilities that are central to embodied agents. Table 15 contrasts REA with prior benchmarks and datasets in terms of task design and evaluation scope.

Embodied-Bench (Yang et al., 2025b) and RoboBrain (Ji et al., 2025) are designed mainly to evaluate VLA models in low-level robotic manipulation and navigation settings, and therefore emphasize a different class of problems than those considered in REA.

Spatial-MLLM (Wu et al., 2025a) and VSI-Bench (Yang et al., 2025a) primarily operate on static-scene video inputs, which are substantially cleaner than real-world embodied observations. Such settings do not reflect the noise and scene perturbations commonly introduced by agent actions in realistic environments (e.g., objects being moved or re-arranged during interaction).

By design, REA is built on real-world, first-person egocentric videos collected during active interactions, capturing natural noise and action-induced changes in the scene. REA further requires models to jointly reason over these egocentric observations and a complete allocentric 3D scene representation, which serves as a cognitive map of the environment and provides a global understanding of the room layout beyond the local video stream. As a result, while egocentric videos only capture the local activity region in front of the agent, REA explicitly evaluates grounding-based spatial reasoning, where egocentric observations are aligned with an allocentric 3D scene representation to enable inference beyond what is directly

*Table 15.* Comparison of Input and Output Modalities Across Datasets.

| Dataset | Input Modalities | Output Modalities |
|---|---|---|
| Embodied-Bench | Video (robot interaction) | Text (high-level / low-level actions) |
| Spatial-MLLM | Video (static scene) | Text (route planning, static object spatial reasoning) |
| RoboBrain | Video (robot interaction) | Text (waypoint trajectories, actions, bbox) |
| VSI-Bench | Video (static scene) | Text (static object spatial reasoning) |
| REA (Ours) | Video (dynamic scene), 3D point-cloud (static) | Text (human motion spatial reasoning) |

*Table 16.* Comparison of training-set scale with related 3D QA benchmarks.

| Dataset | Train | Type |
|---|---|---|
| **REA (ours)** | ∼24k | 3D + ego. video + ST-QA |
| ScanQA | ∼25–30k / 40k | Static 3D QA |
| SQA3D | ∼20k / 33.4k | Static 3D + situated QA |

visible in the egocentric view.

**More broadly, REA is designed to evaluate a model's ability to align multimodal and multi-perspective (e.g., egocentric and exocentric) inputs, while maintaining strong video understanding and reasoning about dynamic spatial relationships over time.** To the best of our knowledge, such grounding-based, multi-perspective spatio-temporal evaluation is not explicitly supported by prior datasets and benchmarks.

**Training-set scale.** REA is comparable in scale to widely used 3D question-answering benchmarks, while covering a more complex spatio-temporal reasoning setting. As shown in Table 16, REA contains about 24k training samples, which is close to the training scale of ScanQA (Azuma et al., 2022) and SQA3D (Ma et al., 2023). Different from these static 3D scene QA datasets, REA pairs egocentric video observations with allocentric 3D point clouds and evaluates joint spatio-temporal reasoning across five task types. This suggests that REA is both practically sized and sufficiently rich for studying multi-modal spatio-temporal reasoning.

### H.3. Dataset Generation Pipeline Runtime

We report the runtime required to generate 5,000 VQA samples for each step in our pipeline, using a single NVIDIA RTX 4090 GPU:

- **Query Video Sampling:**

    - *Relative Direction, Relative Distance:* 1.5 hours
    - *Action Planning, Furniture Affordance, Find My Item:* 3 hours (requires 7B VLM in the loop)

- **3D Position Estimation:** 10 minutes

- **Spatial Relationship Estimation:** 10 minutes

- **Navigation Movement Extraction:** 20 minutes

- **Scene Reconstruction:** 5 seconds per scene, including saving the output as a `.glb` file

- **Frame-to-Point Cloud Registration:** Under 4 hours total for 5,000 query videos (batch size = 1) to retrieve corresponding frames from the database.

## H.4. QA Templates

In this section, we present the QA templates we adopt in our data generation pipeline. Note, the curly-braced values (e.g., {object_1}, {a1}, {direction}) are placeholders. See details in Sec. 3.1.

---

**Relative Direction**

**Single-object:**

**Q:** Does the hand closer to the {object_1} differ when performing '{a1}' and '{ak}'?
**A:** No, the same hand remains closer to the {object_1} during both '{a1}' and '{ak}'.
**A:** Yes, the hand closer to the {object_1} changes from the {direction} of the person to the {direction} of the person between '{a1}' and '{ak}'.

**Q:** Is the {object_1} to the {direction} of the person when the person is performing '{a1}', and {format_direction(direction_at_ak)} when '{ak}'?
**A:** The {object_1} remains {format_direction(direction_at_a1)} during both '{a1}' and '{ak}'.
**A:** Initially, the {object_1} is to the {direction} of the person, but as the person moves, it is to the {direction} of the person.
**A:** At first, the {object_1} appears to the {direction} of the person, but after performing '{ak}', due to the person's movement, it appears to the {direction} of the person.
**A:** Relative to the person, the {object_1} changes from being to the {direction} of the person to the {direction} of the person between '{a1}' and '{ak}'.

**Multi-object:**

**Q:** Are the {object_1} and {object_2} on the same side of the person when performing '{a1}'?
**Q:** Is the person facing both the {anchor_object_1} and {anchor_object_2} from the same side when performing '{ak}'?
**A:** Yes, both the {object_1} and {object_2} are to the {direction} of the person during '{a1}'.
**A:** No, the {object_1} is to the {direction} of the person, while the {object_2} is to the {direction} of the person during '{a1}'.

---

**Relative Distance**

**Single-object:**

**Q:** Does the person move closer to the {object_1} between '{a1}' and '{ak}'?
**Q:** Does the person move away from the {object_1} between '{a1}' and '{ak}'?
**Q:** Does the person end up closer to the {object_1} after performing '{ak}'?
**Q:** Is the person closer to the {object_1} when '{a1}' or when '{ak}'?
**Q:** During which action is the person closest to the {object_1}?
**A:** The person moves closer to the {object_1} from '{a1}' to '{ak}'.
**A:** The person starts off farther from the {object_1} at '{a1}', but ends up closer to it after '{ak}'.
**A:** The person approaches the {object_1} while moving from '{a1}' to '{ak}'.
**A:** The person moves further away from the {object_1} from '{a1}' to '{ak}'.
**A:** The person starts off closer to the {object_1} at '{a1}', but ends up farther from it after '{ak}'.
**A:** The person moves away from the {object_1} while moving from '{a1}' to '{ak}'.

**Multi-object:**

**Q:** During '{a1}', is the person closer to the {object_1} than to the {object_2}?
**Q:** During '{a1}', would it be easier for the person to access the {object_1} or the {object_2}?
**A:** Yes, the person is closer to the {object_1} than to the {object_2} when performing '{a1}'.
**A:** No, the person is closer to the {object_2} than to the {object_1} when performing '{a1}'.
**A:** The person is at a similar distance from both the {object_1} and the {object_2} when performing '{a1}'.
**A:** The person's relative distance to {object_1} and {object_2} is unclear when performing '{a1}'.

**Find My Item**

**Q:** Where is the {object_1}, and how can the person get to it?
**Q:** After performing {action_name}, where did the person leave the {object_1}, and how can it be reached?
**Q:** Would it be closer for the person to bring the {object_1} to the {anchor_object_1} or to the {anchor_object_2}?

**A:** Answers for this task are free-form generations produced by a VideoLLM. Responses may describe the object's location, surrounding context, and suggested navigation steps, depending on the scene and queried action. See Sec. H.5 for more details.

**Furniture Affordance Prediction**

**Q:** Considering the person's previous actions and current movement, which object will they most likely interact with next?
**Q:** Which of the following objects does the person interact with next, given their previous actions and current motion?
**Q:** Based on what the person has done so far and how they are moving now, which nearby object is the person preparing to interact with?

**A:** Answers are free-form generations from a VideoLLM, typically referring to a plausible next object interaction such as "fridge," "microwave," "sink," or "hob." These responses are selected based on the temporal progression and motion cues present in the scene. See Sec. H.5 for more details.

**Action Planning**

**Q:** We are performing a cooking or assembly task with the following sequence of actions: <action_1>, <action_2>, ..., <action_i>. Based on the video, what should I do next, and how can I get to the place where the next step takes place?

**A:** Answers are free-form generations from a VideoLLM, which may describe the predicted next action (e.g., "close salt," "put down spatula") and the spatial guidance for reaching the appropriate location (e.g., "move to the stove," "turn toward the counter on the left"). These answers require reasoning over the temporal context and understanding of task progression. See Sec. H.5 for more details.

## H.5. Answer Generation Prompts

In this section, we present the instructions we use to prompt the VideoLLM to construct free-form ground-truth answers during QA generation.

---

**Find My Item**

**If question_type is "location":**

**<images>** You are given a short video showing the action: {action_name}. In the video, the person places the object: {object_1}.

- The video only shows the **past action** — the moment the object was last placed.

- You, the assistant, are **not in the video**, and the person's current position is unknown.

- You are told the object is now located **{direction_phrase}** from the person's current position. This direction is accurate and must be used in the response.

**Question:** Where is the {object_1}, and how can the person get to it?
**Your answer must:**

- Describe the surroundings around the object at the last moment it was visible (based only on the video)

- Use the known direction "{direction_phrase}" to state where the object is now and how the person can reach it

- Not guess or infer directions from the video

- Not mention the video directly

- Not invent room layouts or paths

- Be one fluent, natural English sentence

**If question_type is "after action":**

**<images>** You are given a short video showing the action: {action_name}. In this video, the person places the object: {object_1}.

- The video shows only the **past action** of placing the object.

- You are **not currently in the video**.

- The person's current position is **after the video ends** and not visible.

- You are told the object is now located at: **{direction_phrase}**, and this direction must be used exactly as given.

**Question:** After performing {action_name}, where did the person leave the {object_1} and how to reach it?
**Your answer must:**

- Describe the surroundings where the object was placed at the end of the action (based only on the video)

- Use the known direction "{direction_phrase}" to describe where the object is now and how the person can reach it

- Not infer direction from the video

- Not use generic phrases like "to the right" unless they match "{direction_phrase}"

- Not mention the video directly or invent room layouts

- Be one fluent and natural English sentence

---

**Furniture Affordance**

**<images>** You are given information about a person and their surroundings:

- **Previous actions performed by the person**: {previous_actions_text}

- **Movement relative to nearby objects**: {movement_text}

- **Available object options**: {options_text}

Your task is to generate a fluent, natural English sentence that answers the following question:
**"Which object will the person most likely interact with next?"**

- The answer should indicate that the person is most likely to interact with the {groundtruth_anchor_object}.

- Do not mention the list of options directly in your answer.

- Explain naturally why the person is approaching or likely to interact with the {groundtruth_anchor_object}, based on their actions and movement.

- Keep the response concise and human-like.

- Do not repeat the question or include unrelated commentary.

**Action Planning**

**<images>** You are an assistant that generates a detailed, natural-language answer in the second person, describing progress and the next step in an egocentric video. The data you have is:

- **Video ID**: {video_id}

- **Video Frames Range**: from {start_frame} to {end_frame}

- **Two key frames for reference**: {", ".join(img_input_list)}

- **Actions in the overall sequence**: {", ".join(all_actions)}

- **Actions completed in the video so far**: {", ".join(completed_actions)}

- **Next action to perform**: {next_action}

- **Motion data for how to perform the next action**: {movement_type}

Your task:

1. Acknowledge which actions have already been completed, based on the video.

2. Infer the user's immediate next step from the provided `next_action`.

3. Describe how the user will physically carry out this next action, considering:

   - Movement type (e.g., forward, backward, left, right, or stand still if movement is minimal)
   - {extra_rotation_instruction}

4. Respond in a single, natural-sounding sentence or short paragraph, addressing the user as "you" (second-person perspective), like an on-the-spot assistant.

5. If the location for the next action is not evident from the video, reference the point cloud to determine where the user needs to go. Then provide navigation instructions (e.g., "move right" or "turn left toward the counter") so the user can reach the correct spot and perform the action.

Please generate a concise, coherent answer that incorporates these details, focusing on telling the user what they have already done and how to perform the next action.

## H.6. Licenses

EPIC-KITCHENS (Damen et al., 2018), EPIC-FIELDS (Tschernezki et al., 2023), VISOR (Darkhalil et al., 2022) are licensed under CC BY-NC 4.0. We thank the authors of these datasets for providing high-quality annotations, which form the basis of our work.

# I. Extended Limitations

For tasks such as *Relative Direction* and *Relative Distance*, we adopt fixed templates to generate candidate answers. While this enables efficient evaluation, it introduces a risk of overfitting to the specific answer formats rather than encouraging diverse and natural responses. To mitigate this, future work could incorporate LLM in the loop to paraphrase or refine the answer templates, promoting more robust generation and improving generalization to non-templated question-answer pairs.

# J. LLM Usage

While preparing this work, we used a large language model (LLM) to assist with language editing. The core research, experimental design, and all scientific claims remain our original work. Beyond editing, LLMs were also employed in the data generation pipeline to generate and refine question–answer pairs, and further served as automated judges for the evaluation of results.

