# OpenReview forum: "Spatio-Temporal LLM: Reasoning about Environments and Actions"
_ICML.cc/2026/Conference — ICML 2026 regular_

### Official Review · Reviewer_WVV4 · 2026-02-20

**Soundness:** 2
**Presentation:** 2
**Significance:** 3
**Originality:** 3
**Overall Recommendation:** 4
**Confidence:** 3

**Summary:**

The paper introduces a framework for Spatio-Temporal reasoning in Multimodal Large Language Models (MLLMs), addressing the gap between global environment understanding (allocentric) and local action observation (egocentric).

**Compliance With Llm Reviewing Policy:**

Affirmed.

**Final Justification:**

Thank you to the authors for the detailed rebuttal. I would keep my initial positive score.

**Key Questions For Authors:**

1. Given the simplicity of the STLLM-3D concatenation approach, did you experiment with more complex temporal-spatial interleaving?
2. How sensitive is the model to point cloud density? Since you reconstructed dense clouds from sparse ones, at what point does "sparse" become "unusable" for effective reasoning?
3. How much does the model’s performance rely on this explicit system prompt vs. learning the modality difference implicitly?
4. How would the current architecture need to change to support tasks where the person moves an object?

**Limitations:**

yes

**Strengths And Weaknesses:**

Strength: The paper correctly identifies that existing benchmarks focus on either static 3D scenes or local video clips, but rarely both. While the individual tools used (Grounded SAM2) are established, the integration into a pipeline that extracts 3D spatial relationships for dynamic actions is a substantial engineering feat.
Weakness:
1. The models primarily rely on concatenating or cross-attending to 3D features without investigating deeper geometric-temporal fusion mechanisms.
2. The pipeline assumes the 3D environment and query objects remain stationary during the video clip. This limits the dataset's applicability to more dynamic real-world scenarios where the environment itself changes.
3. The dataset is restricted to kitchen environments, which may not fully represent the spatial complexity of larger outdoor or multi-room environments.
4. Tasks such as "Relative Direction" and "Relative Distance" are generated using a small set of fixed question-answer templates. This introduces a significant risk that the models are simply learning to "fill in the blanks" of specific linguistic patterns rather than performing deep spatio-temporal reasoning.

---

> ### Author Rebuttal · Authors · 2026-03-31
>
> We thank the reviewer for recognizing the gap our work addresses between static spatial understanding and dynamic temporal understanding benchmarks, and for appreciating the substantial engineering effort behind our data collection pipeline.
>
> QD1: Geometric-temporal fusion exploration:
> > Our two baselines are designed to span a range of fusion complexity: STLLM-3D uses flat concatenation, while STLLM-Aligner employs a cross-modal alignment module with learnable queries attending jointly to point cloud and video features, with an additional variant injecting per-frame geometric cues via camera intrinsics and extrinsics. Finer-grained interleaving, such as per-frame spatial context injection, is a direction we plan to explore in future work.
>
> QD2: Static environment assumption:
> > The current architecture represents the 3D scene as a static global point cloud, which assumes objects remain stationary. To support tasks where a person moves an object, two components would need to change. First, the data pipeline would need to track object positions over time, replacing the static point cloud with a dynamic scene representation that updates as objects are displaced. Second, the model architecture would need to condition on this time-varying scene state, for example, by incorporating per-frame or per-action object-centric 3D representations rather than a single global point cloud. We acknowledge this as an important next step and a natural direction for future work.
>
> QD3: Kitchen-domain scope limitation:
> > As a first step towards evaluating generalization, we perform zero-shot cross-dataset evaluation on SQA3D (Table 3), which covers diverse indoor settings including offices, apartments, and classrooms. Results show that REA training provides some transfer beyond kitchen environments. Extending the REA dataset to larger multi-room and outdoor environments is an interesting direction for future work.
>
> QD4: Fixed template diversity
> > This is discussed in Appendix I. To mitigate this at the evaluation level, we adopt an LLM judge that assesses semantic correctness rather than exact string matching, ensuring that template pattern reproduction alone is insufficient to score well.
>
> QD5: Temporal-spatial interleaving experiments:
> > See QD1 response.
>
> QD6: Point cloud density sensitivity:
> > First, the dense point cloud reconstruction (Sec. 3.1.2) was motivated by pose registration (L198–L202), not model input quality. We did not use the sparse EPIC-FIELDS COLMAP reconstruction for two reasons: (1) COLMAP-based localization of thousands of query video frames is time-consuming at scale, and (2) sparse COLMAP point clouds are too sparse to visually identify objects, making manual quality verification infeasible. Reconstructing dense point clouds with VGGT is fast and produces geometry dense enough for annotators to inspect and verify object positions in our interactive web interface.
>
> > Second, to directly assess sensitivity to point cloud density, we ablate over voxel sizes during downsampling. Results are shown below:
>
> | Voxel Size | # Points (approx.) | Overall Acc. (C/G) | Avg. Acc. (C/G) |
> |---|---|---|---|
> | 0.06 m (paper) | ~6K | 41.89 / 46.50 | 47.32 / 51.29 |
> | 0.12 m | ~3K | 41.50 / 46.80 | 47.30 / 51.39 |
>
> > Performance remains stable across voxel sizes, suggesting the model is robust to point cloud density within this range.
>
> QD7: System prompt vs. implicit modality learning
> > The system prompt is a lightweight positional hint informing the model which tokens encode 3D spatial context vs. egocentric video (Appendix E.1). We expect the model to learn modality differences largely from the data.
> To verify this, we ablate by removing the explicit system prompt entirely, so the model receives no language guidance about which tokens correspond to point cloud or image embeddings. Results are shown below:
>
> | Setting | Overall Acc. (C/G) | Avg. Acc. (C/G) |
> |---|---| ---|
> | With system prompt (paper) | 41.89 / 46.50 | 47.32 / 51.29 |
> | Without system prompt | 40.70 / 44.50 | 46.10 / 49.6 |
>
> > The system prompt provides a modest but consistent improvement (+1.29% C / +2.0% G overall), confirming it offers useful guidance while the model learns modality differences largely from the training signal itself.
>
> QD8: Architecture support for object movements:
> > See QD2 response.

---

> > ### Author Rebuttal · Reviewer_WVV4 · 2026-04-03
> >
> > Thank you for the detailed rebuttal. However, relative performance remains inconclusive, necessitating a more comprehensive comparison with baseline models. I will maintain my current score.

---

> > > ### Author Response · Authors · 2026-04-04
> > >
> > > > We thank the reviewer for the follow-up and would greatly appreciate clarification on which specific comparison the reviewer finds inconclusive. Our results show consistent improvements across two distinct comparison streams in Table 2 (paper):
> > >
> > > - **vs. off-the-shelf VideoLLMs**: STLLM-Aligner (41.89%/46.50%, C/G) outperforms the best off-the-shelf model, VideoLLaMa3 (31.46%/36.03%), by **+10.43%/+10.47%** under both judges.
> > > - **vs. directly finetuned VideoLLMs**: STLLM-Aligner also outperforms the best REA-finetuned VideoLLM, LLaVA-OV-Qwen2-7B† (38.19%/43.65%), by **+3.70%/+2.85%**.
> > >
> > > > Both results demonstrate that the gains are due to the proposed data and the architecture choices of the discussed baselines.
> > >
> > > > To provide an additional comparison, we also evaluate three of the latest off-the-shelf VideoLLMs (QwenVL2.5-7B at 28.00%/34.26%, Qwen3VL-8B at 22.82%/42.00%, and InternVL3.5-8B at 26.98%/36.88%) on our REA data. As shown in the table below, all these models yield results below both STLLM-Aligner and STLLM-3D, confirming that even the latest off-the-shelf VideoLLMs struggle to address spatio-temporal understanding tasks.
> > > > The advantage holds consistently across all five tasks, two independent LLM judges, and zero-shot cross-dataset transfer to SQA3D (Table 3 in the paper). We are happy to provide any additional analysis the reviewer feels is missing.
> > >
> > > | Model | Judge |  Rel. Dir. | Rel. Dist. |  Find Item | Furn. Aff. | Act. Plan. | Overall / Avg. |
> > > | :---- | :---: | ---------- | ---------- | ---------- | ---------- | ---------- | -------------: |
> > > | QwenVL2.5-7B | C |      33.00 |      40.67 |      16.91 |      49.82 |      14.17 | 28.00 / 30.91 |
> > > |  | G |      48.67 |      34.33 |      29.14 |      50.54 |      21.83 | 34.26 / 36.90 |
> > > | Qwen3VL-8B | C |      29.67 |      32.33 |      14.39 |      26.88 |      16.67 | 22.82 / 23.99 |
> > > |  | G |      40.00 |      31.33 |      34.89 |      50.18 |      47.83 | 42.00 / 40.85 |
> > > | InternVL3.5-8B | C |      28.00 |      47.33 |      28.06 |      28.67 |      15.00 | 26.98 / 29.41 |
> > > |  | G |      45.00 |      49.33 |      44.60 |      35.13 |      23.83 | 36.88 / 39.58 |
> > > | *(paper)* |  |            |            |            |            |            |  |
> > > | STLLM-Aligner (7B) | C |      50.67 |      70.67 |      36.69 |      62.72 |      15.83 | 41.89 / 47.32 |
> > > |  | G |      51.33 |      70.67 |      55.04 |      55.56 |      23.83 | 46.50 / 51.29 |
> > > | STLLM-3D (7B) | C |      48.00 |      68.00 |      35.61 |      65.69 |      14.83 | 40.94 / 46.43 |
> > > |  | G |      51.00 |      68.00 |      56.47 |      58.06 |      23.17 | 46.39 / 51.34 |

---

### Official Review · Reviewer_7axT · 2026-03-05

**Soundness:** 2
**Presentation:** 2
**Significance:** 2
**Originality:** 1
**Overall Recommendation:** 3
**Confidence:** 5

**Summary:**

This paper proposes a dataset and benchmark, REA, for studying the spatial and temporal understanding capabilities of MLLMs on reasoning about environments and actions. Correspondingly, the paper provides two baselines, STLLM-3D and STLLM-Aligner, to show the preliminary solutions on the new challenging benchmark. Experimental results show that the new benchmark poses new challenges for current MLLMs, and the proposed baselines show competitive performance.

**Compliance With Llm Reviewing Policy:**

Affirmed.

**Final Justification:**

I would keep my initial score. See the detailed justification in the reply during the discussion phase.

**Key Questions For Authors:**

See weaknesses.

Additional questions:

- How about the upper and lower bounds of the benchmark? For example, what is the human's level? It can also demonstrate the annotation quality.

**Limitations:**

Yes.

**Strengths And Weaknesses:**

Strengths:

- The paper has clear motivations and illustrations, making the idea easy to understand.
- It provides a comprehensive suite for spatial-temporal understanding of MLLMs, from the dataset and benchmark to model baselines.
- The details of the dataset annotation and experiments are clearly presented.
- The proposed baselines achieve new state-of-the-art results on the new benchmark.

Weaknesses:

- Validity of dataset annotations: The annotation pipeline is complicated with many pretrained models in the loop. How to ensure the annotation results are reliable? For example, the reconstruction via the latest feed-forward models may not be precise for the accurate positions or directions. The semantic mask annotations, as well as language annotations, can also be inaccurate. How much human's efforts in the validation procedure?

- The categories of different samples lack enough logic. For example, relative direction and distance can be attributed to spatial understanding. Find my item and action planning can be attributed to temporal understanding. Are there any other questions that are missed in the construction pipeline, and how about the role of "Affordance Prediction"?

- There are many benchmarks regarding spatial-temporal understanding, such as OST-Bench, OmniSpatial, MMSI-Bench/MMSI-Video-Bench. The paper has very little discussion and comparison with these works.

- The dataset scale is also a little limited, given it only has 24k training samples and 1.7k validation samples.

- The baselines compared in Table 2/3 are also limited and out-of-date, lacking the latest MLLMs and 3D-LLMs.

- The analysis in the experiments is of poor writing. For example, the analysis in line 360 does not correspond to the results. The paper is also less than 8 pages long and seems like a rushed version for this submission. More detailed analyses of failure cases and the horizons of different samples are required to understand the new challenges better.

---

> ### Author Rebuttal · Authors · 2026-03-31
>
> Thanks for recognizing our motivation, the comprehensive contribution spanning dataset, benchmark, and strong performance of proposed baselines.
>
> QC1: Annotation pipeline validity:
> > We ensure validity via a three-pronged human effort.
> (1) Point cloud verification: all point clouds are manually inspected via an interactive web interface to confirm object positions relative to the person (Sec. 3.1.2);
> (2) QA pair evaluation: two annotators independently assess 100 test samples across six criteria, with strict consensus required, yielding 81.19% QA quality (Tab. 4);
> (3) LLM judge calibration: judge prompts were iteratively refined against human verdicts until agreement exceeded 97% (App. B.2).
>
> QC2: Task category coherence:
> > Some tasks emphasize spatial or temporal aspects, but all require both by design. Relative direction & distance focus on spatial reasoning, but require temporal grounding. Find my item and action planning emphasize temporal reasoning, but depend on spatial context. Affordance prediction requires to infer motion and predict object interactions by jointly reasoning over video and point cloud data.
>
> QC3: Related benchmark comparison:
> > App. H.2 already includes a dataset comparison (Tabs. 11, 12). We extend this to
>   benchmarks raised by the reviewer:
>
> | Benchmark | Modality | 3D | Training Split | Temporal Action Grounding | Egocentric-Allocentric Alignment |
> |---|---|---|---|---|---|
> | **REA (Ours)** | Ego video + Point Cloud | Yes | Yes (24k samples) | Action-aware | Yes |
> | OST-Bench | RGB-D video | No | Eval-only (10k QA) | No (scene exploration) | No |
> | OmniSpatial | Static images | No | Eval-only (8.4k QA) | No (static images) | No |
> | MMSI-Bench | Static images | No | Eval-only (1k QA) | No | No |
> | MMSI-Video-Bench | video | No | Eval-only (1.1k QA) | Partial | No |
>
> > OST-Bench uses RGB-D streams without a global point cloud. OmniSpatial covers static images with no temporal component. MMSI-Bench/Video-Bench use static images or 2D video without 3D point clouds. REA is the only one requiring joint reasoning over egocentric video and an allocentric 3D scene. We will clarify.
>
> QC4: Training set size:
> > Compared to closely related published benchmarks, the 24k training samples place REA well within the scale of accepted work:
>
> | Dataset | Train Samples | Type |
> |---|---|---|
> | **REA (ours)** | ~24k | 3D point cloud + egocentric video + spatio-temporal QA |
> | ScanQA [1] | ~25–30k (of 40k total) | Static 3D scene QA |
> | SQA3D | ~20k (of 33.4k total) | Static 3D + situated position QA |
>
> > Also, models trained on the 24k REA training data yield strong cross-dataset zero-shot generalization: fine-tuning on REA improves base model accuracy by 2-3% (absolute) on SQA3D (Tab. 3). We believe the combination of scale, multi-task coverage (5 tasks), multi-modal complexity, and demonstrated cross-dataset utility makes REA well-scoped and practical.
>
> > Ref: [1] Azuma et al., "ScanQA: 3D Question Answering for Spatial Scene Understanding," CVPR 2022.
>
> QC5: Recent MLLMs and 3D-LLMs:
> > All models in Tabs. 1-3 are state-of-the-art VideoLLMs. We assess 3 other recent models below. 3D-LLMs are tailored for static scene understanding and not suitable for REA's temporal reasoning.
>
> | Model | Judge |  Rel. Dir. | Rel. Dist. |  Find Item | Furn. Aff. | Act. Plan. | Overall / Avg. |
> | :--- | :---: | --- | --- | --- | --- | --- | ---: |
> | QwenVL2.5-7B | C |      33.00 |      40.67 |      16.91 |      49.82 |      14.17 | 28.00 / 30.91 |
> |  | G |      48.67 |      34.33 |      29.14 |      50.54 |      21.83 | 34.26 / 36.90 |
> | Qwen3VL-8B | C |      29.67 |      32.33 |      14.39 |      26.88 |      16.67 | 22.82 / 23.99 |
> |  | G |      40.00 |      31.33 |      34.89 |      50.18 |      47.83 | 42.00 / 40.85 |
> | InternVL3.5-8B | C |      28.00 |      47.33 |      28.06 |      28.67 |      15.00 | 26.98 / 29.41 |
> |  | G |      45.00 |      49.33 |      44.60 |      35.13 |      23.83 | 36.88 / 39.58 |
> | *(paper)* |  |            |            |            |            |            |  |
> | STLLM-Aligner (7B) | C |      50.67 |      70.67 |      36.69 |      62.72 |      15.83 | 41.89 / 47.32 |
> |  | G |      51.33 |      70.67 |      55.04 |      55.56 |      23.83 | 46.50 / 51.29 |
> | STLLM-3D (7B) | C |      48.00 |      68.00 |      35.61 |      65.69 |      14.83 | 40.94 / 46.43 |
> |  | G |      51.00 |      68.00 |      56.47 |      58.06 |      23.17 | 46.39 / 51.34 |
>
> QC6: L360 and failure cases:
> > The claim on L360 is correct: 31.46% (C) and 39.50% (G) are the best results of any off-the-shelf model in Tab. 2, accurately characterizing the ceiling of current vanilla MLLMs on REA. We will clarify this and add a failure case analysis.
>
> QC7: Human performance:
> > We provide human evaluation of dataset quality in App. B.1 (Tab. 4): expert annotators reviewed samples across multiple criteria, including QA quality, spatio-temporal correctness, and question clarity, confirming strong annotation reliability.

---

> > ### Author Rebuttal · Reviewer_7axT · 2026-04-03
> >
> > The validity of the dataset and benchmark is further demonstrated with more evidence. I would encourage the authors to add more evaluation results and analysis with the latest closed-source MLLMs, such as GPT and Gemini. The evaluation is still limited to a small set of open-source models. More evaluation and analysis can strengthen the validity of the benchmark when pushing the boundary of spatial-temporal understanding capabilities of these foundation models. further supporting the significance of this benchmark.
> >
> > In addition, given that some important content is put into the appendix while the main paper is less than a full 8-page version, I think this paper still needs polishing before being published.
> >
> > Overall, it is a borderline paper given its current stage. I would keep my original rating as a sample point, but I can also accept the paper if other reviewers and ACs think the current results are strong enough and the writing can be polished with a minor revision.

---

> > > ### Author Response · Authors · 2026-04-08
> > >
> > > We thank the reviewer for the constructive feedback. We apologize for the delayed response, as running GPT baseline evaluations required additional time due to data server issues on our end.
> > >
> > > We report GPT-5.2 results as a representative closed-source model below, alongside the latest open-source models evaluated in QC5:
> > >
> > > | Model | Judge |  Rel. Dir. | Rel. Dist. |  Find Item | Furn. Aff. | Act. Plan. | Overall / Avg. |
> > > | :---- | :---: | ---------- | ---------- | ---------- | ---------- | ---------- | -------------: |
> > > | *(open-source)* |  |  |  |  |  |  |  |
> > > | QwenVL2.5-7B | C |      33.00 |      40.67 |      16.91 |      49.82 |      14.17 | 28.00 / 30.91 |
> > > |  | G |      48.67 |      34.33 |      29.14 |      50.54 |      21.83 | 34.26 / 36.90 |
> > > | Qwen3VL-8B | C |      29.67 |      32.33 |      14.39 |      26.88 |      16.67 | 22.82 / 23.99 |
> > > |  | G |      40.00 |      31.33 |      34.89 |      50.18 |      47.83 | 42.00 / 40.85 |
> > > | InternVL3.5-8B | C |      28.00 |      47.33 |      28.06 |      28.67 |      15.00 | 26.98 / 29.41 |
> > > |  | G |      45.00 |      49.33 |      44.60 |      35.13 |      23.83 | 36.88 / 39.58 |
> > > | *(proprietary API)* |  |  |  |  |  |  |  |
> > > | GPT-5.2 | C |      40.33 |      48.00 |      34.89 |      58.06 |      21.17 | 37.05 / 40.49 |
> > > |  | G |      33.33 |      42.00 |      43.88 |      58.42 |      31.33 | 39.78 / 41.79 |
> > > | *(paper)* |  |            |            |            |            |            |  |
> > > | STLLM-Aligner | C |      50.67 |      70.67 |      36.69 |      62.72 |      15.83 | 41.89 / 47.32 |
> > > |  | G |      51.33 |      70.67 |      55.04 |      55.56 |      23.83 | 46.50 / 51.29 |
> > > | STLLM-3D | C |      48.00 |      68.00 |      35.61 |      65.69 |      14.83 | 40.94 / 46.43 |
> > > |  | G |      51.00 |      68.00 |      56.47 |      58.06 |      23.17 | 46.39 / 51.34 |
> > >
> > > GPT-5.2 achieves stronger performance than the open-source models (37.05%/39.78% vs. best open-source 28.00%/42.00%), but remains below our fine-tuned STLLM baselines (41.89%/46.50%). We will include this result in the revised paper to provide a broader view of how REA challenges current models. Gemini API evaluation is not included due to the high cost and reimbursement processing time during the rebuttal period.
> > >
> > > We note that the primary goal of this paper is to propose a new task and training dataset for building multimodal models that jointly reason over 2D egocentric video and 3D point cloud inputs. Evaluating prior 2D-only models’ zero-shot on REA is inherently limited, as these models cannot process the 3D point cloud input and instead rely on multi-view images as a substitute, which does not reflect the intended spatio-temporal reasoning setup.

---

### Official Review · Reviewer_cxW7 · 2026-03-09

**Soundness:** 3
**Presentation:** 3
**Significance:** 3
**Originality:** 3
**Overall Recommendation:** 4
**Confidence:** 3

**Summary:**

This paper attempts to analyze the challenge of joint spatial and temporal reasoning in multimodal large language models (MLLMs), involving egocentric observations and allocentric information. The main contributions of this paper include (1) a novel benchmark REA that evaluates the model's spatial-temporal understanding ability and (2) two baseline models STLLM-3D and STLLM-Align that leverage point cloud to assist spatial QA. Experiments have shown the effectiveness of the models and the generalization ability of the REA dataset.

**Compliance With Llm Reviewing Policy:**

Affirmed.

**Ethical Review Concerns:**

I have read the rebuttal and I would keep my initial positive score.

**Final Justification:**

I would keep my initial positive score.

**Key Questions For Authors:**

Please refer to the Weaknesses.

**Limitations:**

yes

**Strengths And Weaknesses:**

Strengths:
1. This paper brings REA dataset, a novel benchmark that combines video, 3D point clouds, and QA tasks. Five task categories cover different reasoning types, ranging from 3D-aware relative position to action planning.

2. The paper proposes two simple yet effective baselines, a LLaVA-style STLLM-3D model and a BLIP-style STLLM-Aligner, both fusing visual information and 3D point cloud features for robust spatial QA.

3. Experimental results demonstrate 3D point cloud, as auxiliary information, is critical for answering challenging tasks in REA benchmark, outperforming video-only baselines by a large margin.

Weaknesses:

1. The ablation study on the point-cloud branch is not clear enough. As far as I am concerned, the baseline approach should be (1) video input only. (2) video input with additional multi-image input. In the modality integration section, it seems that the baseline approach refers to (2) instead of (1). If so, what is the performance of (1)? Besides, how these multi-view images are sampled?

2. For cross-dataset evaluation, it is encouraged to add results on more spatial understanding benchmarks, e.g. VSI-Bench [1], MMSI-Bench [2], EgoExoBench [3].

[1] Thinking in Space: How Multimodal Large Language Models See, Remember, and Recall Spaces. CVPR 2025.

[2] MMSI-Bench: A Benchmark for Multi-Image Spatial Intelligence. ICLR 2026.

[3] EgoExoBench: A Benchmark for First- and Third-person View Video Understanding in MLLMs. NeurIPS 2025

3. Despite superior performance under LLM judges, the proposed method achieves minor improvement under traditional metrics (Table 1) compared with the baseline LLaVA-Video.

4. In REA benchmark, point cloud and 3D human and object poses are generated automatically, with certain assumptions involved (L187-189, L198-200). Are there any ways to further improve the reliability of these automatically calculated poses and point clouds? Given that the overall quality score is 91\% for the point cloud in Section B.1 (Table 4), this should be nearly 100\% for the test set, otherwise it is not sure whether the incorrect model predictions comes from the inaccurate point cloud input or the model's insufficient spatial-temporal understanding ability.

---

> ### Author Rebuttal · Authors · 2026-03-31
>
> We thank the reviewer for recognizing the novelty of the REA dataset, the effectiveness of our two baselines, and the clear experimental evidence supporting the value of a 3D point cloud as auxiliary information.
>
> QB1: Point cloud ablation:
> > We clarify: the multi-view image baseline (2) is the most natural substitute for the 3D point cloud when evaluating VideoLLM baselines, as existing VideoLLMs cannot directly use point cloud data. We hence provide the 25 multi-view images used for point cloud reconstruction (see Sec. 3.1.2, L209-L217) as a scene-level visual substitute, giving these models the best possible allocentric scene context available without 3D data.
> We additionally provide video-only results below (ChatGPT judge accuracy):
>
> | Model | Rel. Dir. | Rel. Dist. | Find Item | Furn. Aff. | Act. Plan. | Overall |
> |:---|---|---|---|---|---|---|
> | LLaVA-Video-7B-Qwen2 (video only) | 38.3 | 43.3 | 18.7 | 52.0 | 10.0 | 28.6 |
> | STLLM-Aligner (video only) | 43.3 | 55.0 | 30.9 | 65.2 | 13.3 | 36.6 |
> | *(paper)* | | | | | | |
> | STLLM-Aligner | 50.7 | 70.7 | 36.7 | 62.7 | 15.8 | 41.9 |
> | STLLM-3D | 48.0 | 68.0 | 35.6 | 65.7 | 14.8 | 40.9 |
>
> > Removing the point cloud branch causes a 5.3% overall drop (41.9 to 36.6). The degradation is most pronounced on spatially demanding tasks: Relative Distance drops by 15.7% and Relative Direction by 7.4%. This is expected as these tasks require precise global scene geometry that video alone cannot provide. Find My Item also drops by 5.8%, as locating out-of-view objects depends heavily on allocentric scene context. This confirms that the point cloud branch provides complementary spatial information beyond what can be inferred from egocentric video frames.
>
> QB2: Cross-dataset evaluation:
> > We note that these benchmarks operate under fundamentally different paradigms than REA. VSI-Bench [1] includes a training split (VSI-Train-10k) and is designed for fine-tuning evaluation rather than zero-shot transfer. EgoExoBench [3] is designed to evaluate cross-view semantic alignment between paired egocentric and exocentric perspectives, which is a different capability from REA's spatial grounding within a 3D scene. Evaluating our model, which is trained solely on egocentric video with 3D context, would not constitute a meaningful test on this benchmark. MMSI-Bench [2] is the most suitable for zero-shot cross-dataset evaluation. We report results below. *All values are accuracy (%). Bold within a column indicates the better result between the base model and STLLM-Aligner.*
>
> |  | Pos. Rel. | | | | | | Attr. | | Motion | | MSR | **Avg.** |
> |---|---|---|---|---|---|---|---|---|---|---|---|---|
> | | Cam.-Cam. | Obj.-Obj. | Reg.-Reg. | Cam.-Obj. | Obj.-Reg. | Cam.-Reg. | Meas. | Appr. | Cam. | Obj. | | |
> | LLaVA-Video-7B-Qwen2 (Base) | 28.0 | **30.9** | 23.5 | **36.0** | **40.0** | **33.7** | 25.0 | **34.8** | **14.9** | **28.9** | **23.7** | **28.6** |
> | STLLM-Aligner | **36.6** | 27.7 | **25.9** | 33.7 | 32.9 | 32.5 | **26.6** | 27.3 | 12.2 | **28.9** | 20.7 | 27.2 |
>
> > The overall slight drop (28.6% to 27.2%) is expected given the paradigm mismatch: MMSI-Bench uses static multi-image inputs without 3D point clouds, whereas our model is fine-tuned to jointly reason over egocentric video and an allocentric 3D scene. **Appearance (-7.5%)** drops because REA does not train visual attribute recognition, suppressing appearance-focused features in favor of geometric reasoning. **Object-Region (-7.1%)** drops because REA is egocentric-focused, reasoning about objects relative to the person's viewpoint rather than semantic scene regions. **Cam.-Cam. (+8.6%)** improves as REA trains with per-frame camera intrinsics and extrinsics, which transfers to viewpoint-to-viewpoint spatial reasoning.
>
> QB3: Limited gains under traditional metrics:
> > Improvements under traditional metrics are indeed smaller. However, REA is very sensitive to fine-grained semantic differences, such as left versus right: a small wording change can alter correctness. These nuances are not well captured by traditional metrics, which focus more on surface-level similarity. In contrast, LLM based judges better reflect semantic correctness, making them more suitable for evaluating spatio-temporal reasoning in our setting.
>
> QB4: Annotation quality concern:
> > While the overall point cloud score of 91% (Table 4) is not perfect, it remains substantially higher than the current model accuracy, making it unlikely to be the primary bottleneck. The remaining issues are mainly due to challenges during reconstruction, such as fast camera motion or motion blur. We manually filter and verify reconstructed scenes to mitigate these issues, but some edge cases remain. We agree that further improving reconstruction quality is valuable, and this could be achieved with more robust multi-view reconstruction or additional filtering in future work.

---

> > ### Author Rebuttal · Reviewer_cxW7 · 2026-04-05
> >
> > Thanks for the authors effort. The rebuttal addressed my concerns and I would keep my initial positive score.

---

> > > ### Author Response · Authors · 2026-04-08
> > >
> > > We thank the reviewer for the thorough and positive review. We are glad the rebuttal addressed the concerns. We will add the video-only ablation results and zero-shot MMSI-Bench evaluation in the revised version.

---

### Official Review · Reviewer_uqcE · 2026-03-12

**Soundness:** 3
**Presentation:** 3
**Significance:** 2
**Originality:** 3
**Overall Recommendation:** 4
**Confidence:** 4

**Summary:**

This paper studies a largely underexplored setting in which answering questions requires joint reasoning over a global 3D environment and a local egocentric action video. It introduces the REA dataset, which contains five task types, and proposes two baselines, STLLM-3D and STLLM-Aligner, that take multiple modalities as input to improve spatial understanding. Experiments show that off-the-shelf MLLMs perform poorly on REA, while the proposed STLLM variants achieve clearly better results and also improve performance on existing benchmarks such as SQA3D.

**Compliance With Llm Reviewing Policy:**

Affirmed.

**Final Justification:**

The authors have addressed all my concerns and I will maintain my positive score.

**Key Questions For Authors:**

How well do the proposed models generalize to other indoor domains with different patterns such as layouts and object categories?

Can the authors quantify the trade-off between computation and accuracy of the proposed STLLM-3D and STLLM-Aligner and provide some analysis?

**Limitations:**

Yes

**Strengths And Weaknesses:**

Soundness.
The paper is technically solid overall. The dataset generation pipeline is carefully designed and grounded in existing data sources. Robustness is further strengthened by techniques such as manual validation and prompt refinement. The modality analysis showing that replacing 3D point clouds with 2D multi-view inputs leads to a performance drop also supports the paper’s claim. However, the empirical gains are still moderate in absolute terms, and the strongest evidence remains benchmark-based rather than grounded in embodied evaluation such as in-door navigation. Another weakness is that the paper explicitly uses dynamic video together with a static point-cloud map, while objects may move during interaction, potentially leading to scene inconsistency.

Presentation.
The paper is clearly organized and easy to follow. The motivation is well stated, the figures are informative, and details such as training hyperparameters and prompt design are clearly presented in the appendix. One suggestion is that the contributions in the introduction could be further refined and made more specific.

Significance.
The research problem is important: some settings require grounding a local stream of observations in a broader world model, while existing benchmarks do not fully capture this scenario. In this sense, the REA dataset itself is a valuable contribution, even beyond the specific baselines proposed in the paper. The cross-dataset gains on SQA3D further suggest that the benchmark teaches nontrivial transferable capabilities. However, one limitation is that the benchmark mainly focuses on kitchen-like egocentric videos, while generalization to other environments, such as offices, remains unexplored.

Originality.
The main motivation of holistic spatio-temporal understanding is novel, but the proposed model, which fuses multiple modalities including 3D, is commonly seen in related work.

---

> ### Author Rebuttal · Authors · 2026-03-31
>
> We thank the reviewer for the constructive review, recognizing the technical soundness of our pipeline, the clarity of our presentation, and the significance of REA as a dataset contribution.
>
> QA1: Empirical gains:
> > We respectfully disagree that the gains are moderate. When replacing multi-view images with point clouds, performance improves from 36.99%/ 42.92% (C/G)  to 40.94%/46.39% in overall accuracy. This corresponds to a consistent improvement of about 4% absolute across both metrics, which is nontrivial given the difficulty of the REA benchmark.
>
> QA2: Evaluation on indoor navigation:
> > Great suggestion. We agree that embodied evaluation, such as indoor navigation, is exciting. At the same time, such evaluation requires substantial system-level integration beyond the scope of this paper. Our current focus is to isolate and study spatio-temporal reasoning in a controlled benchmark setting. Indoor navigation is an important and natural next step.
>
> QA3: Objects moving during interaction:
> > Thanks for raising this insightful point. In our current dataset, object movements are small and localized, and do not significantly alter the global scene structure. For this reason, we adopt a static point cloud representation, which provides a stable global spatial context while the egocentric video captures short-term interactions. We agree that modeling fully dynamic scenes is important, and incorporating dynamic point clouds is a natural and valuable extension.
>
> QA4: Refine contributions in introduction:
> > Great suggestion. We will revise the contribution statements as follows: (1) for the dataset, we will specify the scale (24k training samples), the 5 task categories, and the unique property of requiring joint egocentric-allocentric reasoning over video and 3D point clouds; (2) for the baselines, we will clarify the architectural distinction between STLLM-3D (direct concatenation) and STLLM-Aligner (cross-modal alignment), and highlight their cross-dataset generalization on SQA3D.
>
> QA5: Dataset scope and significance:
> > As discussed in the paper (L292-L308), we start with kitchen scenes because they already provide rich interactions, diverse objects, and complex activities, making them a strong and practical testbed for spatio-temporal reasoning. We did explore expanding to broader domains, but currently, there are limited large-scale datasets that provide both egocentric video and aligned 3D structure. In particular, we attempted to use Ego-Exo4D [1], but its API has been unavailable for several months. We view extending REA to more diverse environments as an important direction for future work.
>
> > [1] Grauman, Kristen, et al. "Ego-exo4d: Understanding skilled human activity from first-and third-person perspectives." Proceedings of the IEEE/CVF Conference on Computer Vision and Pattern Recognition. 2024.
>
> QA6: Model novelty concerns:
> > We would like to clarify that our focus is on establishing the problem and dataset. The proposed models serve as initial baselines for this new setting. While some architectural components are related to prior work, the overall design differs in order to jointly interpret global spatial structure from a point cloud and temporal dynamics from egocentric video to answer a textual query. We view these baselines as a strong starting point, and expect that they can be further improved in future work.
>
> QA7: Generalization to other indoor domains:
> > We evaluate generalization via cross-dataset transfer on SQA3D (Tab. 3 in the paper), which covers 650 scenes and 1,603 test QA pairs across offices, apartments, and classrooms with 40 object categories. Despite this domain gap from REA's kitchen-centric setting, STLLM-3D reaches 55.21%/55.65% (C/G) and STLLM-Aligner reaches 54.40%/54.71% (C/G), both outperforming all off-the-shelf baselines in Tab. 3.
>
> QA8: Computation-accuracy tradeoff between STLLM baselines:
> > Great question! Both models use similar sequence lengths at the LLM decoder, but differ in how spatial information is processed. STLLM-3D directly uses point cloud features and is more computationally efficient. STLLM-Aligner introduces an additional alignment module, which increases computation. In terms of performance, STLLM-Aligner shows only marginal improvement over STLLM-3D, indicating that STLLM-3D offers a simple and efficient alternative with competitive performance. We’ll add to the revised paper.

---

> > ### Author Rebuttal · Reviewer_uqcE · 2026-04-04
> >
> > My concerns have been fully resolved. Hope the authors can add the computational trade-off analysis in the revised version. I will keep my weak accept score.

---

> > > ### Author Response · Authors · 2026-04-08
> > >
> > > We thank the reviewer for the thorough and positive review. We are glad the rebuttal addressed the concerns. We will include the computational trade-off analysis comparing STLLM-3D and STLLM-Aligner in the revised version.

---

### Decision · Program_Chairs · 2026-04-30

**Decision:**

Accept (regular)

**Comment:**

The paper considers equipping LLMs with spatio-temporal awareness for solving real-world tasks. To this end, it introduces the 'Reasoning about Environments and Actions' (REA) dataset, combining existing datasets including EPIC-Kitchens for ego-centric videos, EPIC-FIELDS for 3D kitchen environments, and VISOR for object segmentations, totaling about 25k samples. The paper presents five tasks on the dataset, including relative distances and directions capturing the proximity of an actor’s whereabouts, finding items, furniture affordance, and action planning. An STLLM is proposed, extending the LLaVa-Video-7B-Qwen2 model through cross-modal alignment and fine-tuning for reasoning over the combined modalities. Experiments demonstrate about a 10% improvement of STLLM-3D over prior non-fine-tuned models on this dataset.

The paper received mostly positive reviews, with one weak reject recommendation. Reviewers generally appreciated the attempt to address a largely underexplored setting of reasoning over both global 3D scenes and local egocentric videos. They also raised several issues, including:

1. moderate empirical gains of the proposed method (uqcE), and the need for stronger ablation studies and cross-dataset evaluations (cxW7, 7axT)
2. the focus of the dataset on kitchen scenes alone, without sufficient study of generalization to other settings (uqcE, WVV4)
3. lack of compelling motivation for the designed tasks on the REA dataset (7axT, WVV4)
4. outdated baselines (7axT)

The authors provided a rebuttal with clarifications addressing these concerns:

1. Ablation studies show consistent performance improvements from the chosen modalities, along with additional experiments using video-only inputs.
2. While acknowledging limitations in generalization, the authors provide a small-scale study on SQA3D demonstrating promising transfer.
3. The authors contrast REA with prior datasets, clarifying key differences.
4. Updated results include GPT-5.2 achieving 39% overall performance versus 46% for STLLM-3D.

The AC independently reviewed the paper and concurs with the reviewers that the proposed REA dataset addresses an important problem of spatio-temporal reasoning jointly over 3D point clouds and temporally localized video scenes, and could be valuable for advancing research in this emerging area. While the proposed STLLM-3D is somewhat incremental, the empirical results demonstrate state-of-the-art performance. Overall, the AC believes the paper makes a solid contribution and recommends acceptance. The authors should incorporate the reviewers’ feedback and the additional results in the revised version.